# Decision-making flexibility in New Caledonian crows, young children and adult humans in a multi-dimensional tool-use task

**Rachael Miller**[1]☉*, **Romana Gruber**[2]☉, **Anna Frohnwieser**[1]*, **Martina Schiestl**[2,3], **Sarah A. Jelbert**[4], **Russell D. Gray**[2,3], **Markus Boeckle**[1], **Alex H. Taylor**[2], **Nicola S. Clayton**[1]

**1** Department of Psychology, University of Cambridge, Cambridge, England, United Kingdom, **2** School of Psychology, University of Auckland, Auckland Central, New Zealand, **3** Max Planck Institute for the Science of Human History, Max Planck Society, Jena, Germany, **4** School of Psychological Science, University of Bristol, Bristol, England, United Kingdom

☉ These authors contributed equally to this work.
\* rmam3@cam.ac.uk (RM); af643@cam.ac.uk (AF)

**Data Availability Statement:** The full data set is available on Figshare: https://figshare.com/s/8e1670783219ddc4561e

## Abstract

The ability to make profitable decisions in natural foraging contexts may be influenced by an additional requirement of tool-use, due to increased levels of relational complexity and additional work-effort imposed by tool-use, compared with simply choosing between an immediate and delayed food item. We examined the flexibility for making the most profitable decisions in a multi-dimensional tool-use task, involving different apparatuses, tools and rewards of varying quality, in 3-5-year-old children, adult humans and tool-making New Caledonian crows (*Corvus moneduloides*). We also compared our results to previous studies on habitually tool-making orangutans (*Pongo abelii*) and non-tool-making Goffin's cockatoos (*Cacatua goffiniana*). Adult humans, cockatoos and crows, but not children and orangutans, did not select a tool when it was not necessary, which was the more profitable choice in this situation. Adult humans, orangutans and cockatoos, but not crows and children, were able to refrain from selecting non-functional tools. By contrast, the birds, but not the primates tested, struggled to attend to multiple variables—where two apparatuses, two tools and two reward qualities were presented simultaneously—without extended experience. These findings indicate: (1) in a similar manner to humans and orangutans, New Caledonian crows and Goffin's cockatoos can flexibly make profitable decisions in some decision-making tool-use tasks, though the birds may struggle when tasks become more complex; (2) children and orangutans may have a bias to use tools in situations where adults and other tool-making species do not.

## Introduction

Effective decision-making ensures that individuals achieve goal-directed behaviour [1–3]. In natural foraging contexts, individuals are required to take into account various different aspects simultaneously when making profitable decisions, such as whether to travel further afield for higher quality foods and how they can access extractable foods, for instance, through the use of tools. Such decisions may therefore be influenced by work-effort sensitivity, the level

**Funding:** The study was funded by the European Research Council under the European Union's Seventh Framework Programme (FP7/2007-2013)/ ERC Grant Agreement No. 3399933, awarded to N. S.C (PI). R.G., M.S., A.H.T. received funding from a Royal Society of New Zealand Rutherford Discovery Fellowship and a Prime Ministers McDarmid Emerging Scientist prize awarded to A. H.T. The funders had no role in study design, data collection and analysis, decision to publish, or preparation of the manuscript.

**Competing interests:** The authors have declared that no competing interests exist.

of perceived risk, attention to the functionality of available tools, and the quality of food available [4, 5].

One aspect that underlies decision-making is self-control—the capacity to suppress immediate drives in favour of delayed rewards [6]. One approach to measuring self-control is the use of delay of gratification tasks, where subjects have to wait longer and/or work harder to obtain a more valuable outcome [7]. In human children, delay of gratification is developmentally influenced, shows high individual variation and correlates with measures of success in later life, such as social and academic competence [8], though see a recent study [9]. Self-control skills emerge in infancy [10, 11] and develop throughout toddlerhood and pre-school age [12, 13], improving significantly between ages 3 and 5 [14, 15] and further beyond 5 years.

Delayed gratification is also likely important for non-human species in a number of contexts, including foraging and social interactions. A prominent example is tool-use, where animals may have to forgo immediate gratification when they choose to use a tool to gain access to high value but out-of-reach food, rather than low value, freely accessible food. One possibility, therefore, which is predicted by the technical intelligence hypothesis [16], is that self-control is enhanced in tool-using species compared to non-tool using species. Specifically, tool-using species may have evolved better self-control abilities so they can make more efficient decisions when foraging with their tools. However, evidence for this hypothesis is mixed.

The ability to forgo an immediate reward for a delayed one has been found in various non-human animals, including primates and birds—see [17] for a recent review of self-control in crows, parrots and non-human primates—though has primarily been tested in non-tool-using contexts. To date, no clear comparisons can be drawn between the self-control abilities of tool-using and non-tool-using species, also based on the fact that different methodologies have been used for different species. For example, in tasks not involving tool-use, where subjects choose between two food qualities, one available immediately and one following a delay, tool-using capuchin monkeys (*Sapajus apella*) performed comparably to tool-using great apes (*Pan paniscus*, *Pan troglodyes*), and outperformed non-tool-using marmosets (*Callithrix jacchus*) and tamarins (*Saguinus oedipus*) [18, 19]. However, non-tool-using spider monkeys (*Ateles geoffroyi*) actually outperform capuchins [20]. While some species are able to use and/or make tools in the lab, for instance, Goffin's cockatoos [21, 22], there is limited evidence that they do so in the wild [23]. Therefore, we refer to these species as 'non-tool-making'. A number of non-tool-making species, including carrion crows (*Corvus corone*), common ravens (*Corvus corax*) and Goffin's cockatoos, perform similarly to tool-making chimpanzees at non-tool-using delay tasks [24–27]. Furthermore, non-tool-making Eurasian jays (*Garrulus glandarius*) and California scrub-jays (*Aphelocoma californica*) are able to overcome current motivational needs in non-tool-use contexts; i.e. during caching and when food-sharing with a partner [28–30].

Tasks involving delayed gratification in a tool-use context have been tested in separate single-species studies, primarily using a variety of paradigms, in tool-making primate species (e.g. chimpanzees [31], capuchins [32], orangutans [5]) and in non-tool-making bird species (Goffin's cockatoo [33] and common ravens [34]). In these studies, the choice is typically between getting a lesser reward without a tool versus using a tool to gain a better reward. Experience with tool-use improves performance in a delayed gratification tool-use task in capuchin monkeys, where subjects could either immediately consume rod-shaped food items or carry them to an apparatus to use them to extract a food of high quality [32]. Chimpanzees and orangutans selected a tool over a least preferred food and range of toys when the tool could be used later to suck fruit soup from a bowl [31]. Non-tool-making bird species have also shown delayed gratification ability in a tool-use context. Ravens were able to consistently select the correct tool over distractor items including an immediate reward to open a box and obtain a reward, even when the box was missing for up to 17 hours [34], though see [35, 36]. Goffin's cockatoos were

able to overcome immediate drives in favour of future gains in performance on a delayed gratification tool-use task [33].

However, very few studies have used comparable methodology with both tool-making and non-tool-making species in tool-using tasks involving delayed gratification, with a focus on the ability to flexibly make the most profitable decision. Additionally, benchmarking against children and adult human performances is crucial, given that past tests of physical cognition have found differences between human and non-human species' performances, which suggest that the task may not measure the ability that it is designed to test, or only measures it in one species but not the other. This can be due to differences in perceptual abilities or assumptions about human performance. One prominent example of this is the trap-tube task [37–41], in which subjects have to push or pull food from a horizontal tube while avoiding a trap. When humans were faced with the inverted trap tube, where the trap was now on top of the horizontal tube and therefore inactive, they failed to avoid the trap [42]. This task had been previously presented as a key test of causal understanding in humans and other species. Thus, researchers should not assume that failure in a task indicates poorer performance of one species over another, without having tested both species using the same (or at least closely comparable) methodology.

Here, we tested children aged 3–5 years, adult humans and tool-making New Caledonian crows [43, 44] on a multi-dimensional tool-use task, requiring the use of two different types of tools, apparatuses and rewards varying in quality as determined by a preference test. We focused on the ability to make the most profitable decisions across five conditions where reward quality, tool functionality and work effort were manipulated. We adapted and extended an experimental paradigm previously tested on non-tool-making Goffin's cockatoos [33] and tool-making orangutans [5]. Subjects were required to make binary choices between two tools (stick or stone) or a tool and a reward (most or least preferred) to use in one of two apparatuses. We refer to the previous findings in cockatoos and orangutans, in addition to the present tested species, though note some minor differences in methodology between species, as outlined in the methods and discussion. We can therefore make tentative comparisons between species, with a primary focus on exploring each species' ability to make profitable decisions in multiple contexts that each requires use of a tool. We expected that–similar to cockatoos and orangutans [5, 33]–New Caledonian crows and humans would be able to show flexibility in their ability to make profitable decisions in this multi-dimensional paradigm. Additionally, delayed gratification and tool functionality understanding appear to develop in children at different ages and increase with age [45, 46], but have not previously been tested simultaneously. Thus, we expected children's ability to solve these tasks to increase with age. Therefore, we were also able to provide novel insight into the developmental trajectory and inter-relation between these abilities in children.

## Materials and methods

### Ethics statement

The methods of this study were carried out in accordance with the relevant guidelines and regulations. The study and related experimental protocols were approved and conducted under the European Research Council Executive Agency Ethics Team (application: 339993-CAUS-COG-ERR) and University of Cambridge Psychology Research Ethics Committee (pre. 2013.109). Informed written consent was obtained from legal guardians prior to child participation, and from adult subjects. The parents of the children identified in the Supplementary Video gave their informed written consent for this information to be published. The New Caledonian crow research was conducted under approval from the University of Auckland

Animal Ethics Committee (reference number 001823) and from the Province Sud with permission to work on Grande Terre, New Caledonia, and to capture and release crows.

While care was taken to make the methodology as similar as possible between the humans and crows, please note that there were some differences in methodology between species as outlined in the methods and discussion sections, which limits the opportunity to directly compare the species with one another [47]. We therefore focus primarily on individual species performance in these tasks.

## Subjects

**New Caledonian crows.**   Bird subjects were 6 New Caledonian crows caught from the wild (at location 21.67˚S 165.68˚E) on Grand Terre, New Caledonia. The birds were held temporarily in captivity on Grande Terre for non-invasive behavioural research purposes from April to August 2017. There were 4 males and 2 females, based on sexual size dimorphism [48], of which 4 were adults and 2 were juveniles (less than 1 year old), based on age by beak colouration [48]. Due to the small number of birds held at the field site at any time, it was not possible to include a larger sample, and juvenile and adult crows were grouped for the analysis. The birds were housed in small groups, consisting of two to four individuals per group, in a ten-compartment outdoor aviary, with approximately 7x4x3m per compartment, containing a range of natural enrichment materials, like logs, branches, sea shells and pine cones. Subjects were tested individually in temporary visual isolation from the group. The birds were not generally food deprived, and the daily diet consisted of meat, dog food, eggs and fruit, with water available *ad libitum*. The birds were trained to stand on weighing scales for a small food reward to regularly monitor their weight, and all birds maintained at or above capture weights during their stay in captivity. The birds were acclimatized to the aviaries in April and trained for the experiment in May–July. All birds completed the full study in August 2017. At the end of their research participation, birds were released at their capture site(s). A previous study indicated that New Caledonian crows housed temporarily in a similar situation as the present study successfully reintegrated into the wild after release [49].

**Children and adult humans.**   Child subjects were 88 children aged between three and five years old: 29 3-year olds (Mean: 3.69 years; Range: 3.31–3.93 years), 30 4-year olds (M: 4.45 years; R: 4–4.96 years) and 29 5-year olds (M: 5.42 years; R: 5–5.93 years), of which 46 were male and 42 were female. We chose the age range of 3–5 years for children as previous research suggests significant improvements in self-control ability within this range [14, 15]. Children were recruited and tested at five nurseries and primary schools in England, serving predominantly white, middle-class communities, between January and February 2018. 20 adult subjects were recruited via the Cambridge psychology research sign-up system and tested in April 2018, comprising primarily of Undergraduate and Postgraduate students at the University of Cambridge, 3 were male and 17 were female. Adults received £5 for participation in the study. All adults and children tested completed the full study. Humans were tested individually in temporary visual isolation, though for some of the very young children, a member of staff was present in the room, but did not interact with the child.

## Materials

The 'stone-apparatus' was a box made of Perspex (10x5cm) with a vertical tube on top (8x3cm) and a platform inside that collapsed when a heavy object, i.e., a stone, was dropped (Fig 1). To prevent subjects from inserting the stick into the stone-apparatus to release the platform and get access to the reward, the vertical tube had a 30˚ slant, which made the release of the platform with a stick impossible. We used two different 'stick-apparatuses' for the humans

## Stone apparatus                          Stick apparatuses

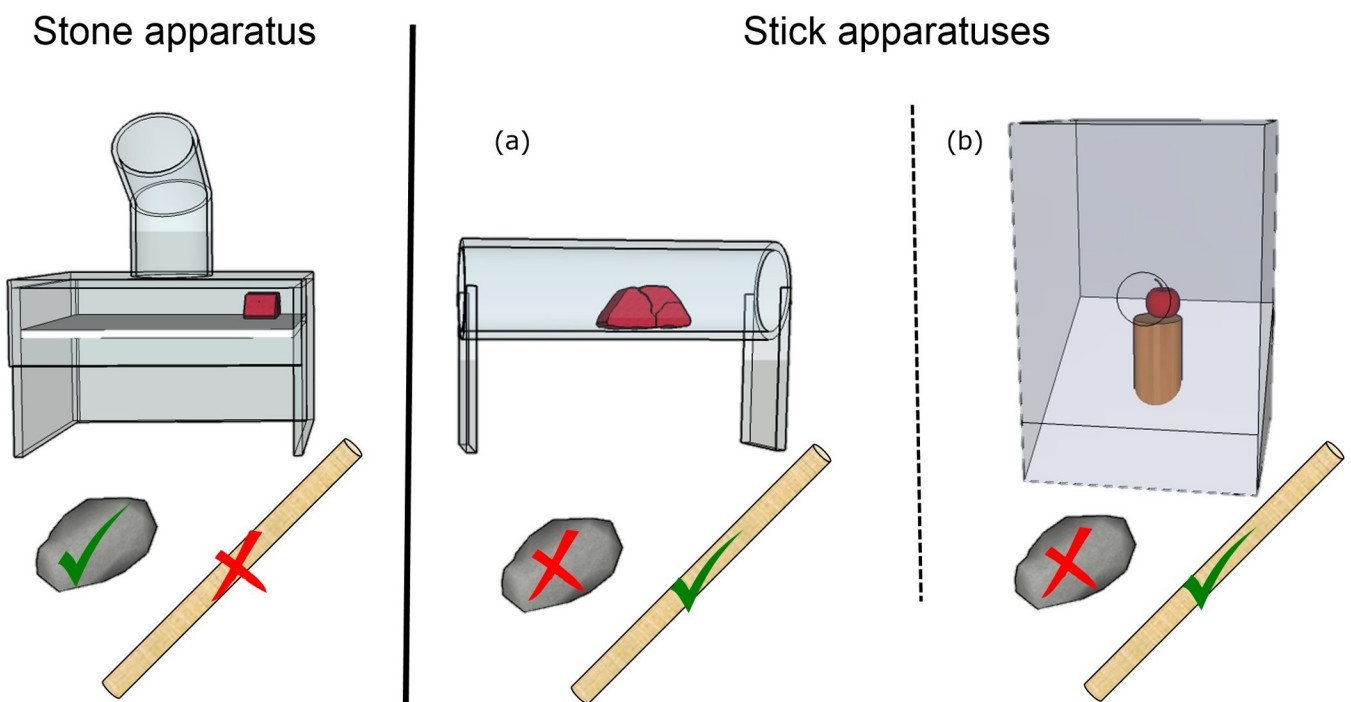

**Fig 1.** Stone-apparatus (left) and stick-apparatus (right), a) shows the stick-apparatus for the crows and b) for the humans, with both tools—the functional tool indicated by a green hook and the non-functional tool with a red cross.

and the crows though both apparatuses were functionally the same, as they required a stick to contact a reward and move it to the left or right. The minor variation in the apparatus structure was due to the testing equipment available in New Caledonia. The 'stick-apparatus' for the humans was similar to [50] with a transparent box with a central opening hole (17x17cm), where the reward sits on a small platform on the slanted plate within the box. For the crows, we used a 'stick-apparatus' consisting of a horizontally orientated Perspex tube (10x3cm) that rested on two Perspex pillars (5cm high), which could be operated by a wooden stick to gain access to the reward by raking the reward towards themselves causing it to drop in to the subject's reach (Fig 1A and 1B). To prevent the subject from inserting the stone into the stick-apparatus to try to obtain the reward, the entrance hole was too small and narrow on both apparatus types for stone insertion. Therefore, only the stone was functional in the stone-apparatus and only the stick was functional in the stick-apparatus.

The crow experiment was run in a similar manner to the Goffin's cockatoo study [50], using the same apparatuses, tools and protocol for training and testing, in order to enable comparisons of performance in each bird species. However, we made further adaptations to the crow study, by extending the previous study, as detailed below. The human experiments were run as closely as possible to the bird experiments, using the same apparatuses, tools and protocols. Fewer trials were run for the humans than birds due to practical reasons like restrictions on session length and number for the children. The reward types also differed between species and groups. The rewards used for the crows were meat as the most preferred food reward and a piece of apple as the least preferred food reward, following reward preference testing. The crow rewards differed from the cockatoos (nuts), due to the differing diets of these two species. The child rewards were three types of stickers of similar sizes: most preferred (animal stickers), least preferred (white, square stickers) and 'medium' stickers (round, yellow, smiley face

stickers) as determined by the reward preference test and piloting. The reward for the adults was money (£), with 10p as least preferred, 50p as most preferred and 20p as medium reward, represented by white, square, laminated tokens of the same size as the stickers used for the children.

### New Caledonian crow experiment

**Procedure.** During training and testing, the subjects were presented with a binary choice between two tools or a tool and a reward, on the left and right side of the apparatus, with the side of presentation semi-randomly balanced across trials.

**Training.** Unlike the cockatoos in the previous study, the crows were wild-caught and not comfortable with close human presence, therefore we adapted the methodology slightly for the crows. The choice for training and testing was presented inside two of five drawers resting on a table and could be operated by the experimenter behind a visual barrier (See S2 Video). The crows could sit on an elevated perch diagonally to the rear of the table and a perch in front of the drawers, which allowed them to inspect the contents from a distance and close up, pick up the tools and operate the apparatuses. Depending on the condition, either one apparatus was placed in front of the middle drawer, and the tools were presented in the drawers left and right of it, resting on a piece of foam so that the crow could easily pick it up, or both apparatuses were placed left and right of the middle drawer, and the tools were presented in the middle drawer. The drawers were pulled back if the crow either successfully received the reward or made a wrong connection between tool and apparatus. They were never pulled back when the subject was operating the apparatus to avoid disturbing the subject.

There were two steps to the training phase. In Step 1, the subject had the opportunity to learn tool use, until they could reliably retrieve the reward. The crows received training to drop stones into the stone-apparatus as per previous studies using this apparatus (e.g. [51, 52]. After each bird had dropped the stone into the tube 20 times without any mistakes, they moved on to the next training step. As the crows were natural tool users, they did not require any pre-training for stick use, though were habituated to the stick-apparatus and had to retrieve the reward from the apparatus 20 times.

In Step 2, we checked that the reward quality preferences were viewed as such by the subject in a reward preference test, with 11 sessions of 12 trials each until the subject selected the most preferred reward over the least preferred one in 80% of binary choices. 9 sessions were run prior to testing, one session during testing prior to running the *tool selection quality allocation* condition, and a final session after all tests were completed. Subjects were presented with similar sized pieces of meat, bread, dog food and apple. Although subjects showed minor individual preferences between meat, bread and dog food, all subjects consistently selected these items over apple. Hence, meat was selected as the most preferred reward and apple as the least preferred reward for all birds.

**Testing.** In the *tool selection* condition, the subject should select the functional tool from the choice of both tools–one functional and one non-functional to the presented apparatus–to obtain the reward inside the apparatus (Fig 2A). In the *motivation* condition, the subject should avoid work effort by selecting the immediately available most preferred reward. The choice was between the functional tool and most preferred reward, with the apparatus containing the exact same most preferred reward (Fig 2B). In the *quality allocation* condition, the subject should select the functional tool over the immediately available least preferred reward, but take the immediately available most preferred reward. For each apparatus, the choice was between the functional tool and least preferred reward with the most preferred reward inside the apparatus, or between the tool and most preferred reward with the least preferred reward

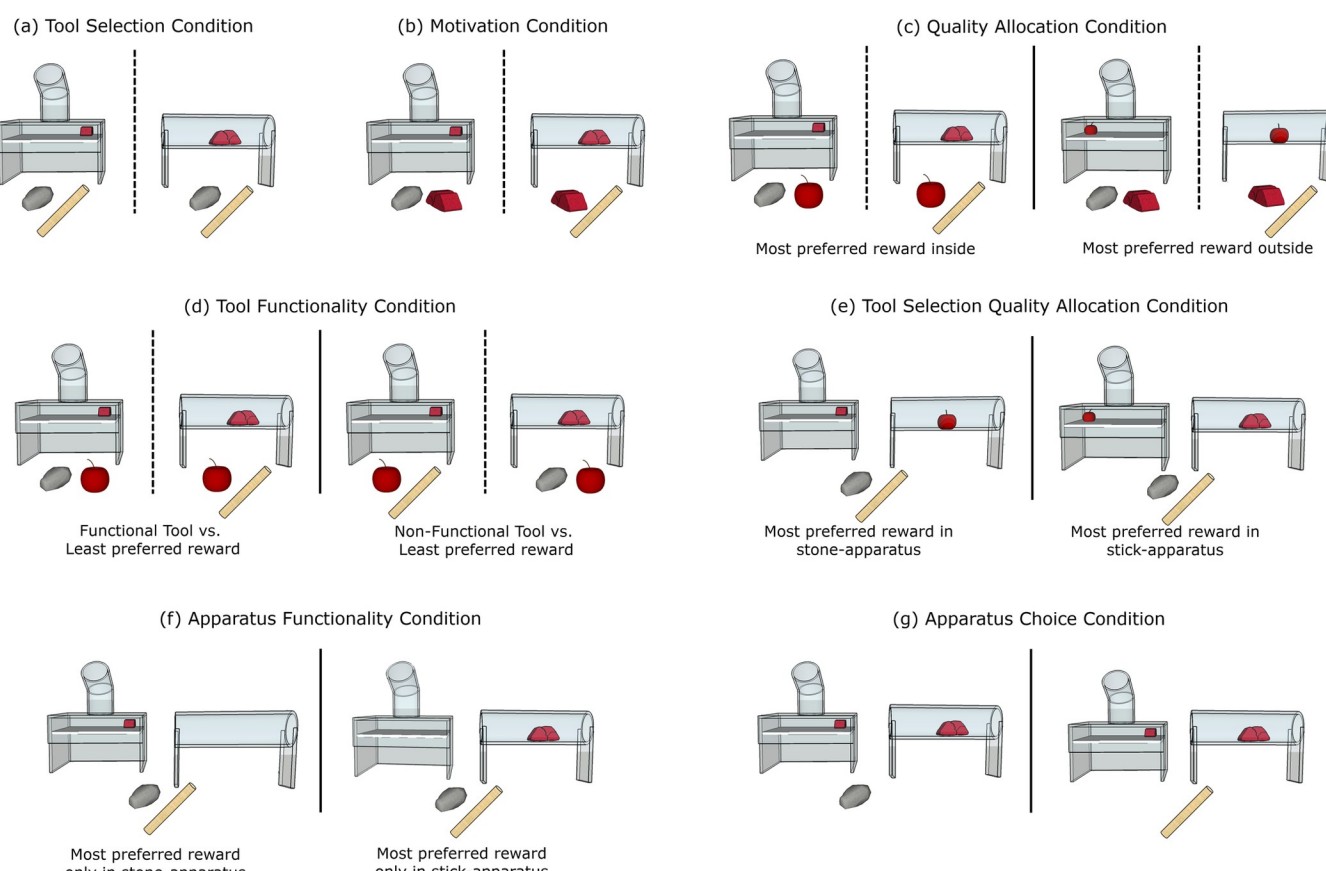

**Fig 2. All conditions.** (a) Tool Selection Condition: both tools present, most preferred food (MPF) inside: (b) Motivation Condition: functional tool present, MPF inside and outside; (c) Quality Allocation Condition: functional tool present, either MPF inside (left), and the least preferred food (LPF) outside, or MPF outside and LPF inside (right); (d) Tool Functionality Condition: functional tool present (left) or non-functional tool present (right), MPF inside and LPF outside; (e) Tool Selection Quality Allocation Condition: both apparatuses present with both tools, MPF in stone-apparatus (left), or stick-apparatus (right), LPF in other apparatus; (f) Apparatus Functionality Condition: both apparatuses and both tools present, MPF in stone- (left) or stick- (right) apparatus, other apparatus empty; (g) Apparatus Choice Condition: only one (functional) tool is present, both apparatuses presented and baited with MPF.

inside the apparatus (Fig 2C). In the *tool functionality* condition, subjects should select the tool over the least preferred reward only when the tool was functional. For each apparatus, the choice was between the functional tool and the least preferred reward or the non-functional tool and the least preferred reward, with the most preferred reward inside the apparatus in both cases (Fig 2D). In the *tool selection quality allocation* condition, subjects should select the functional tool for the appropriate apparatus that contained the most preferred reward. In this condition, all task components were present, with both tools present, and the most preferred reward either in the stone-apparatus and the least preferred reward in the stick-apparatus, or the other way around (Fig 2E).

In the *apparatus functionality* condition, we explored whether crows could choose the correct tool for the correct apparatus, by presenting subjects with both apparatuses and both tools, but only one apparatus was baited with most preferred food, while the other apparatus was empty (Fig 2F). In the *apparatus choice* condition, subjects should choose the correct apparatus for the available tool, with both apparatuses presented and baited with most preferred food, though only one tool was present (Fig 2G). The *apparatus functionality* and *apparatus choice* conditions were new additions to the [50] cockatoo study. These new conditions were included for the crows, as both bird species struggled with the tool selection quality

allocation condition, so we aimed to explore whether giving more experience when both apparatuses were present at the same time, and further sessions of the *tool selection quality allocation* condition, could improve the crows' performance in this final condition.

## Children and adult humans experiments

**Training.** We did not use the drawers for the human study to present the choices, but rather presented the choices on the table in front of the subject, in the same way that the choice was presented to the cockatoos. The human training was the same as the crows, except the humans received fewer trials per learning step than the crows. Specifically, in step 1, the human subject was shown how to use the tool and then could try for 1 trial per apparatus. In step 2, the subject could try to use the non-functional tool for up to 10 seconds, before it was replaced with the functional tool, which they could use to obtain the reward in 1 trial per apparatus. In step 3, three reward preference trials were run (1 trial = most vs. less preferred reward, 1 trial = tissue vs most preferred reward, 1 trial = tissue vs less preferred reward) with an additional trial run at the end of the test session (most vs. less preferred reward) to confirm that this key preference still held. In step 4, eight trials were run with the apparatus containing a medium preferred reward and the choice between the functional tool and a piece of tissue, or the apparatus containing the piece of tissue and the choice between tool and medium preferred reward.

The human experiment also included a verbal command during training–"you can have the immediately available item now or the tool to try to use later", randomising which item was mentioned first between trials. During all trials, if the subject chose the tool over the immediately available item in any trial, they had to wait before they were allowed to use it as the experimenter pulled the apparatus back out of the subject's reach for 5 seconds and then pushed it back into reach and said 'go', whether their choice was correct or incorrect. We included this command in the human experiment during piloting, after we discovered that the children preferred to select the tool in the motivation condition, where the choice was between the functional tool and the immediately available most preferred reward, with the exact same most preferred reward inside the apparatus. We aimed to explore whether this command may incur a small cost of selecting the tool for the humans.

**Testing.** The human test procedure was the same as the crow one, other than reducing the trials per condition for the humans (see below).

**Crows and humans: Test trials.** For the crows, in *tool selection*, *apparatus functionality* and *apparatus choice* conditions, the crows received a minimum of 2 sessions of 12 trials each until 18 of 24 trials were correct in 2 consecutive sessions. In *motivation* and *tool selection quality allocation* conditions, the crows received 2 sessions of 12 trials each per condition. In *quality allocation* and *tool functionality* conditions, the crows received 4 sessions of 12 trials each per condition. The incorrect choice in each condition resulted in no reward (*tool selection*, *tool functionality*, *apparatus functionality*, *apparatus choice* conditions), least preferred reward (*tool selection quality allocation*, *quality allocation*, *tool functionality* conditions) or the most preferred reward (*motivation* condition). Within every condition, the trials were randomized across session and individuals. For the humans, in the *tool selection*, *motivation* and *tool selection quality allocation* conditions, humans received two trials (one per apparatus) per condition. In the *quality allocation* and *tool functionality* conditions, humans received four trials per condition (two per apparatus). We did not run the additional two conditions on apparatus functionality and apparatus choice with humans, as, unlike the birds, piloting indicated that humans did not struggle with the *tool selection quality allocation* condition.

**Crows and humans: Test order.** The crows and humans were each divided into 2 different subgroups to control for learning effects, each subgroup receiving a different order of

condition: tool selection–motivation–tool functionality–quality allocation–tool selection quality allocation, the second group received the following order of tool selection–quality allocation–motivation–tool functionality–tool selection quality allocation. All crows then received the same order of the additional tests designed to further explore performance in the tool selection quality allocation condition: another 2 sessions of the *tool selection quality allocation— apparatus functionality–apparatus choice–tool selection quality allocation* (sessions 5–8).

## Data analysis

We recorded the choice per trial for each subject as 'correct' or 'incorrect'. All test sessions were coded live as well as being video-recorded (unless parental consent requested otherwise for the children). 10% of trials, i.e. 240 trials for the crow data and 242 trials for the human data, were coded from video by a second observer and compared to the live coding, finding significant agreement for the human data ($\kappa = 1$; *p = < .001)* and the crow data ($\kappa = .818$; *p = < .001)*. Example trials can be found in S1 and S2 Videos.

We conducted Generalized Linear Mixed Models (GLMM: [53] using R (version 2.15.0; R Core Team, 2014) using the R packages lme4 [54] and nlme [55] (for GLMM), MASS [56] (allows for negative binomial), gamlss [57] (beta distributed data) and multcomp [58] (posthoc tests within glmm) to assess which factors influenced success rate in the children and New Caledonian crows. Success was a binary variable indicating whether the subject correctly solved the trial (1) or not (0) and was entered as a dependent variable in the models. For the crows, we included the random effect of subject ID and the random slopes for trial and condition within subject ID, fixed effects of condition (1–5), apparatus type (stick/stone), gender (male/female), trial number (1–12) and age (adult/juvenile). When the problem of non-convergence occurred, the complexity of the model was reduced by dropping single factors separately until convergence was reached while keeping maximum complexity of the model. The resulting model included condition, sex, trial, age, and (1 + trial|ID) as random effect. For the children, we included the random effect of 1+Trial+Condition|ID, fixed effects of age in decimal years (continuous: ages 3–5 in individual years), condition (1–5), gender (male/female), trial number (1–14) and the interaction between age and condition. When the problem of non-convergence occurred, the complexity of the model was reduced by dropping single factors separately until convergence was reached while keeping maximum complexity of the model. The resulting model included condition, gender, trial, age, and (1 | ID) as random effect. We used deviance information criteria to compare the full model (all predictor variables, random effects) firstly with a null model, and then with reduced models to test each of the effects of interest [59]. The null model consisted of random effects and no predictor variables. The reduced model comprised of all effects present in the full model, except the effect of interest [59]. Posthoc comparison of factors in the model exceeding two categories were calculated with Tukey correction for multiple comparisons.

For the crows, we then analysed the data in a comparable way to the cockatoo and orangutan data, using non-parametric two-tailed statistics, namely 1-sample Wilcoxon tests and Mann-Whitney U tests run in SPSS version 21. The cockatoo and orangutan data were obtained from the Supplementary Materials in [5, 50]. For the children, we ran further analyses using exact two-tailed Binomial tests to assess success rate in each condition, for each apparatus type separately, and age class (3–5 years). We focussed on group-level analyses, in order to ease any comparisons between species, however, the individual-level analyses for the crows, using two-tailed Binomial tests, can be found in S1 Table, with crow subject information in S2 Table.

## Results

### New Caledonian crows

For the crow data, the full model differed significantly from the null model ($\chi^2$ = 198.67, df = 94, $p = < .001$). We found a significant main effect of condition (Estimate = -3.580, $\chi^2$ = -6.005, $p = < .001$) on success rate (correct vs. incorrect choice, S3 and S4 Tables for multiple comparisons of the factor condition). The crows generally performed well in the *tool selection*, *motivation* and *quality allocation* conditions, and performed poorly in the *tool functionality* and *tool selection quality allocation* conditions (Table 1). Specifically, in the *tool selection* condition, when choosing the correct tool for the presented apparatus, the crows chose correctly significantly above chance with both apparatus types combined (Table 1) and the stick-apparatus alone, though not with the stone-apparatus alone. In the *motivation* condition and *quality allocation* condition, the crows chose correctly significantly above chance level with both apparatus types combined (Table 1) and the stick- and stone- apparatus alone.

In the *tool functionality* condition, when the correct choice was either the least preferred reward over the non-functional tool or the functional tool over the least preferred reward, the crows did not select correctly significantly above chance, either with both apparatus types combined (Table 1), nor with the stick- or stone- apparatus alone. In the *tool selection quality allocation* condition, when all task components were present at once, looking at session 1 and 2 only, the crows did not select correctly significantly above chance with both apparatus types combined (Table 1), nor with each apparatus type alone.

After the first two sessions, unlike in the cockatoo study [50], the crows in the present study were given a further two sessions to see whether additional experience would improve performance. In these two further sessions of the *tool selection quality allocation* condition, the crows selected correctly above chance with both apparatus types combined (Table 1) and with the stone-apparatus, but not with the stick-apparatus alone. Following this, again unlike the cockatoos, the crows then received further experience of both apparatuses being presented at once in the apparatus functionality and apparatus choice conditions. In the *apparatus functionality* condition, where both apparatuses and tools were present but only one apparatus was baited, the crows selected correctly significant above chance with both apparatus types combined and the stone-apparatus alone, though not the stick-apparatus alone. In the *apparatus choice* condition, where both apparatuses were baited and presented with only one tool, the crows selected correctly significantly above chance (Table 1). Following these two additional conditions, the crows received four additional sessions of the *tool selection quality allocation* condition. Across

**Table 1. Performance across all conditions for the crows across both apparatuses.** Results reflect results of Wilcoxon 1-sample signed ranks tests–chance value = 50%. Significant p-values (< .05) highlighted in bold. S1-8 stands for the number of sessions in this condition.

| Condition | T$^+$ | P |
|---|---|---|
| Tool selection (S1&2) | 21 | **.026** |
| Motivation | 21 | **.02** |
| Quality Allocation | 21 | **.027** |
| Tool functionality | 21 | .223 |
| Tool selection quality allocation (S1&2) | 19 | .074 |
| Tool selection quality allocation (S3&4) | 21 | **.027** |
| Tool selection quality allocation (S5-8) | 21 | **.027** |
| Apparatus functionality | 21 | **.028** |
| Apparatus choice | 21 | **.027** |

sessions 5–8, the crows selected correctly significantly above chance with both apparatus types combined and singly (Table 1).

## Children and adult humans

For the child data, the full model differed significantly from the null model ($\chi^2 = 259.1$, df = 114, $p \leq .001$). We found a significant interaction effect of age and the *tool functionality* condition ($\chi^2 = 2.469$, $p = 0.014$) as well as the *motivation* condition ($\chi^2 = -2.463$, $p = 0.014$) on success rate in children (correct vs. incorrect choice; S5 Table). Success rate increased with age in the *tool functionality* and *motivation* conditions and was also significantly poorer in both of these conditions for all ages, compared with the other conditions.

When combining all conditions, only the 4 and 5-year old children selected correctly above chance across all trials, while the 3-year olds did not select correctly above chance (Table 2). Within conditions, the 3-year olds did not select correctly above chance in any condition, except for the *tool selection quality allocation* condition (Table 2). The 4-year olds and 5-year olds performed well and comparably to one another, selecting correctly above chance within the *tool selection*, *quality allocation* and *tool selection quality allocation* conditions (Table 2). 3 to 5-year olds showed a non-significant trend to select *incorrectly* on the *motivation* condition–i.e. to select the tool even though the reward immediately available and inside the apparatus were exactly the same (both rewards were most preferred). 3 to 5-year olds did not select correctly above chance in the *tool functionality* condition. In adults, we found that subjects selected the correct choice above chance across all conditions and within each condition separately (Table 2).

In the *tool functionality* condition for children, 3-, 4- and 5-year old children selected significantly above chance when the tool presented was functional, i.e. they chose the functional tool over the least preferred reward (Table 3). When the tool presented was non-functional, 3 to 5-year olds significantly selected *incorrectly* above chance, i.e. they incorrectly chose the non-functional tool over the least preferred reward, while adults chose correctly (Table 3). In the *quality allocation* condition for children, 5-year olds selected correctly significantly above chance in all trials. The 3- and 4-year olds selected correctly significantly above chance when the most preferred reward was inside the apparatus and the correct choice was the tool, but when the least preferred reward was inside the apparatus, so the correct choice was the immediately available most preferred reward, 3-year olds did not have a significant preference for the correct choice and 4-year olds chose correctly with the stick apparatus but not the stone apparatus (Table 3). Adults chose correctly in all conditions (Table 3).

**Table 2. Correct choices (%) in tool selection, tool selection quality allocation and motivation conditions within each apparatus type for each age (3–5 years) for children and for adult humans.** P-values calculated from exact two-tailed binomial tests. Significant p-values are highlighted in bold. NS = not significant with a Bonferroni correction. "Incorrect"–selected incorrect choice above chance.

| Age (in years)/ Apparatus type | Tool selection | | Tool selection quality allocation | | Motivation | |
|---|---|---|---|---|---|---|
| | % | *p* | % | *p* | % | *p* |
| 3 –stone | 68 | *.071* | 72 | ***.024*** | 32 | *.07* |
| 3 –stick | 81 | ***.001*** | 86 | ***< .001*** | 26 | ***.011 incorrect*** |
| 4 –stone | 87 | ***< .001*** | 83 | ***< .001*** | 27 | ***.016 incorrect*** |
| 4 –stick | 100 | ***< .001*** | 83 | ***< .001*** | 37 | *.2* |
| 5 –stone | 90 | ***< .001*** | 97 | ***< .001*** | 24 | ***.008 incorrect*** |
| 5 –stick | 97 | ***< .001*** | 93 | ***< .001*** | 34 | *.14* |
| Adult–stone | 100 | ***< .001*** | 100 | ***< .001*** | 80 | ***.005*** |
| Adult–stick | 100 | ***< .001*** | 100 | ***< .001*** | 85 | ***.003*** |

**Table 3. Correct choices (%) in tool functionality and quality allocation conditions in each apparatus type for children aged 3–5 years and adult humans.** P-values calculated from exact two-tailed binomial tests. NS = not significant with a Bonferroni correction. Significant p-values are highlighted in bold. "Incorrect"–selected incorrect choice above chance.

| Age (in years)/ apparatus type | Tool functionality | | | | Quality allocation | | | |
| --- | --- | --- | --- | --- | --- | --- | --- | --- |
| | Functional choice | | Non-functional choice | | Least preferred reward inside apparatus | | Most preferred reward inside apparatus | |
| | % | *p* | % | *p* | % | *p* | % | *p* |
| 3—stick | 90 | *< .001* | 20 | *.001 incorrect* | 47 | *.856* | 87 | *< .001* |
| 3—stone | 93 | *< .001* | 13 | *< .001 incorrect* | 57 | *.585* | 90 | *< .001* |
| 4 –stick | 90 | *< .001* | 27 | *.016 incorrect* | 73 | *.016* | 93 | *< .001* |
| 4—stone | 87 | *< .001* | 20 | *.001 incorrect* | 67 | *.099* | 90 | *< .001* |
| 5—stick | 90 | *< .001* | 28 | *.024 incorrect* | 86 | *< .001* | 97 | *< .001* |
| 5 –stone | 90 | *< .001* | 72 | *.024 incorrect* | 90 | *< .001* | 93 | *< .001* |
| Adult—stick | 100 | *< .001* | 100 | *< .001* | 100 | *< .001* | 100 | *< .001* |
| Adult—stone | 100 | *< .001* | 95 | *< .001* | 100 | *< .001* | 100 | *< .001* |

## Comparison with previous studies in Goffin's cockatoos and orangutans

We compare the performance of young children, adult humans and tool-making New Caledonian crows tested in the present study, with that of tool-making orangutans [5] and non-tool making Goffin's cockatoos [50] tested in previous studies. As there were minor methodological differences between species, and therefore necessary differences in analyses, we make only tentative species comparisons, rather focusing on performance per species as above. We focus on performance across apparatus types (stick- and stone- apparatus) rather than separately, though present the crow and cockatoo comparisons for each apparatus type in S6 and S7 Tables. We found that adult humans selected correctly above chance in all conditions. Crows, cockatoos, orangutans and children aged 4–5 selected correctly significantly above chance in the *tool selection* and *quality allocation* conditions. Children aged 3–5 and orangutans performed poorly in the *motivation* condition, while crows and cockatoos performed well. Children aged 3–5 and crows performed poorly in the *tool functionality* condition, while cockatoos and orangutans performed well. Children aged 3–5 and orangutans performed well in the *tool selection quality allocation* condition, while crows and cockatoos performed poorly in this condition (Fig 3).

## Discussion

Our study was designed to investigate the ability of tool-making New Caledonian crows, 3-5-year-old children and adult humans to make profitable decisions requiring effective decision-making, delayed gratification and work-effort sensitivity in a tool-use context. We found that these species performed in a similar manner to non-tool-making Goffin's cockatoos [50] and tool-making orangutans [5] tested in previous studies. They were able to flexibly select between reward items of differing quality and tools (including non-functional tools) relative to the context of each condition. As there were differences between the two species in training, reward types and apparatuses, we cannot compare species directly, and so focus on both species' performances in both experiments.

First, we found that the crows and cockatoos performed well in most conditions, except the *tool functionality* (cockatoos only) and *tool selection quality allocation* conditions (crows and cockatoos). In this latter condition, all task components were present and the most profitable choice was the tool for the apparatus containing high-quality food, while ignoring the other tool for the second apparatus containing low-quality food. The orangutans and humans

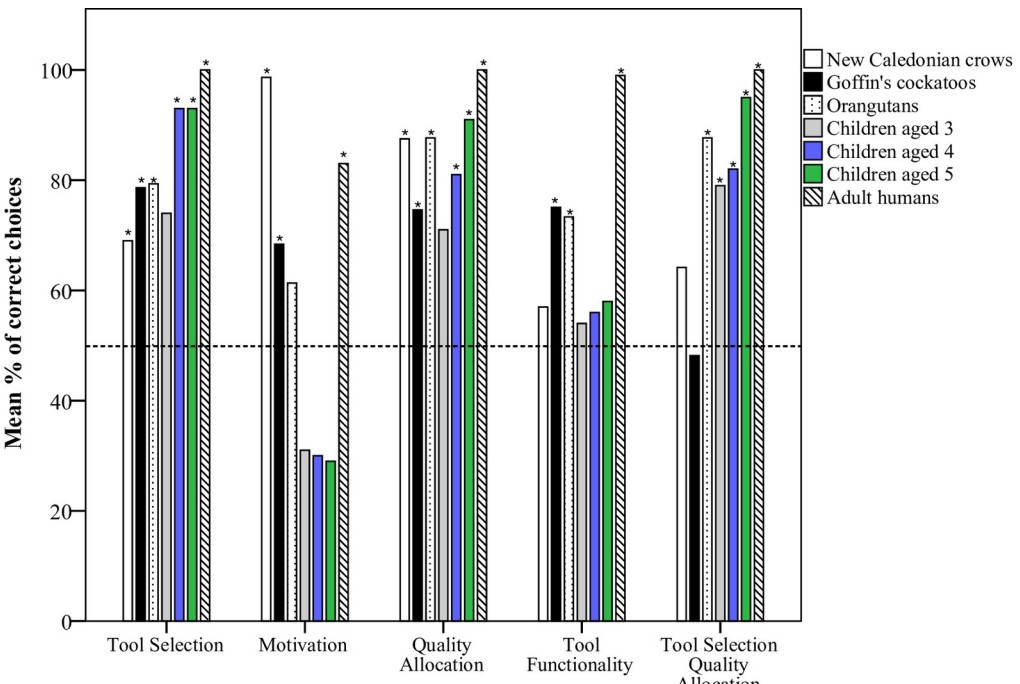

**Fig 3. Mean percentage of correct trials for each condition for New Caledonian crows, children aged 3–5 years, adult humans (present study), Goffin's cockatoos [50] and orangutans [5].** * indicate significant selection of correct choice within condition for each age from exact two-tailed binomial tests for the children and adults, and 1-sample Wilcoxon signed rank tests for the crows, cockatoos and orangutans. Dashed horizontal line indicates chance level.

performed significantly above chance, while the birds did not, in this condition. However, additional experience (not available in the cockatoo study) improved performance for the crows in this condition.

Second, the birds and adult humans performed significantly above chance, while the 3–5 year old children and orangutans did not, in the *motivation* condition, where the most profitable choice was to take the preferred reward that was immediately available on the table, rather than use a tool to obtain the same preferred reward from inside an apparatus, indicating work-effort sensitivity. Third, the cockatoos, orangutans and adult humans performed significantly above chance, while the crows and children did not, in the *tool functionality* condition, where the most profitable choice was to take an immediately available reward rather than try to use a non-functional tool, which would not result in a reward, indicating an ability to take into account tool functionality while making decisions. Specifically, children and crows often chose the non-functional tool over an immediately available reward.

That 3-year old children struggled with most problems requiring the ability to delay gratification and in understanding how to succeed on various tasks echoes past child development work, such as [14, 46]. However, our results also suggest that experience may improve tool-related delayed gratification performance, given that they passed the *tool selection quality allocation* condition. The birds' performance in the four tasks that the 3-year olds struggled with, and particularly the *motivation* condition, which 4-5-year olds and orangutans also struggled with due to an apparent preference for using tools, is particularly intriguing, though note that the birds received more trials in this condition than children. Generally, these performances indicate that these tasks are non-trivial in terms of motivation and the cognition required and so illustrate the high levels of self-control that these species possess, even when solving tool

problems. The *motivation* condition, where children and orangutans continued to select the tool even when the same high-quality reward was immediately available, suggests that they, though not the adult humans or birds, may have a bias towards tool-use. That is, when all things are equal, our result suggests children and orangutans prefer to use a tool to get a reward over the more efficient choice of directly taking an equal reward. However, once tool-use leads to different types of food, as in the *quality allocation* condition, orangutans and 4-5-year olds, though not 3-year olds, appear able to overcome the desire to immediately select the tool. This could be caused by an inability to inhibit taking a tool before evaluating the reward options or a different understanding of the goal of the task, i.e. exploring the tools and apparatuses instead of focussing on acquiring the rewards.

Another possibility is that there may have been an issue of contra-freeloading, i.e. that the use of the tool was a reward in itself. In many trials, children tried to use the incorrect tool to acquire the reward inside the apparatus and were given a chance to do so until they gave up; however, some children also realised their mistake after making the wrong choice and confirmed so verbally without trying to use the non-functional tool or when receiving a non-preferred reward. It is unusual that contra-freeloading would influence the children and orangutans, though not the other groups tested. As the crows were wild birds, unlike the cockatoos and orangutans, they may have had more of a food-shortage mentality, i.e. be less likely to contra-freeload. However, a recent study by McCoy et al. [60] showed that after tool-use, New Caledonian crows are more optimistic towards a task compared with when they had not used tools, which shows that tool-use in itself is rewarding for the crows, and so does not provide support for the contra-freeloading hypothesis in relation to this study. Similarly, the cockatoos–which were captive birds—were able to refrain from selecting the tool when it was not necessary. As children aged 3–5 continued to select the tool even when it was not required, rather than just the youngest children, this suggests that it is not an issue with understanding of needing a tool if one is available. Further work is clearly needed to investigate the possibility of a bias towards using tools that even other tool-making species, such as New Caledonian crows, do not have. Such a bias could potentially explain differences between the tool behaviours of great apes and other species, just as a bias in the motivation to cooperate may explain differences between chimpanzee and human cooperation [61]. When given a choice between working together or alone for the same reward, children, though not chimpanzees, show a preference for working together [61], though recently one kea also showed this bias [62].

In a surprising result, both crows and children, though not the cockatoos, orangutans or adult humans, performed poorly in the *tool functionality* condition, as they selected the non-functional tool over the immediate reward. These results cannot be explained by a tool functionality understanding issue, given these subjects reliably discriminated between functional and non-functional tools in the *tool selection* task. There are a number of other potential hypotheses. First, crows and children may have a drive to 'give it a go' and try to make non-functional tools work, rather than accept a low-value reward, due to past experience that sometimes tools that appear not to work can be made to function. Thus, both these subject groups may have higher persistence than cockatoos, who have less experience of such situations. However, the success of human adults and orangutans counts against this hypothesis, as they would likely have as much, if not more experience, that non-functional tools sometimes can be made to work.

One possibility is potential species differences in the perceived reward value associated with each particular test. The rewards did differ between subject groups, as they were selected to be most appropriate and desirable for each group. Hence, the crows and children may have selected the non-functional tool over the least preferred reward due to the low preferential value assigned to the latter type. The difference between food rewards for the cockatoos was most preferred and third (least) preferred reward. In comparison, the crows were presented

with meat as most preferred reward and apple as least preferred reward, as they showed consistent preference for meat over apple. However, the quality of the rewards used for the other subject groups also differed from one another, yet they selected correctly in this condition. A final possibility is that these results are directly due to differences in self-control. Despite the large amounts of experience crows and children had in deciding when to use a tool and when a tool was not functional, these subjects may have struggled to inhibit picking up a non-functional, previously rewarded tool in this task. As described above, in the *motivation* condition, children continued to select a (functional) tool over a high-quality reward, indicating that tool-use itself may have been rewarding for them, which could have been the case even when the tool was non-functional. Cockatoos, orangutans and adults in contrast, may simply have had sufficient self-control to ignore this tool. Future work incorporating both tool-related and non-tool-use delayed gratification paradigms across species are required to further explore this possibility.

The clearest difference in performance between the primates and birds appeared in the *tool selection quality allocation* condition. Here, both the crows and cockatoos struggled, while orangutans, 3-to 5-year old children and adult humans performed well. This was also the only condition that 3-year old children passed. The authors in a previous study [50] suggested that the poor cockatoo performance in this condition reflected possible information processing limitations, with subjects being unable to focus their attention on all relevant cues at once [63]. Additionally, chimpanzees, bonobos and orangutans struggled when facing a trap-tube problem requiring simultaneously considering two spatial object-object relations (e.g. tool-reward), which the authors suggested may be due to cognitive overload in the attentional system [64]. In the present study, we found that the crows also struggled with this condition, at least within the first two sessions, though three individuals did perform above chance. We included two new conditions for the crows that were not used in the cockatoo study [50], where the subjects gained more experience of selecting the correct tool or apparatus when both apparatuses were present, before further testing in the *tool selection quality allocation* condition. We found that this additional experience did improve crow performance in this condition. Our findings support the suggestion by [50]: without extended experience, birds, though not great apes–at least in this context, though not necessarily in others [64]—may have issues in attending to multiple variables at once. This may affect problem-solving performance in other physical cognition tasks.

New Caledonian crows and Goffin's cockatoos show comparable performance in some cognitive tasks, such as tool-manufacture in the lab and flexible problem-solving skills [21, 22, 38, 44, 65, 66]. They both show high levels of object manipulation in and outside the foraging context [65] and make intrinsically structured object combinations [67]. New Caledonian crows often use stick tools in the wild, which may explain why they performed better using the stick-apparatus than the stone-apparatus in the *tool selection* condition. However, it does not explain why they performed better with the stone-apparatus than the stick-apparatus in the *apparatus functionality* condition. Importantly though, only the crows routinely make and use tools in the wild [43], while there is limited evidence that cockatoos make tools in the wild [23], though they can do so proficiently in the lab [22].

Despite this, we found that the crows performed similarly to the cockatoos. This result is in line with studies indicating that non-tool-making ravens perform comparably with tool-making chimpanzees in a delayed gratification tool-use context [34]. It also corresponds with previous findings that New Caledonian crows do not have higher levels of motor self-regulation–defined as stopping a pre-potent but counter-productive movement–than non-tool making carrion crows [68]. Clearly, there are several potential explanations for why the cockatoos performed well in the *tool functionality* condition, while the crows struggled, including different experiences (cockatoos were hand reared with extensive testing experience, while the crows were wild-caught), minor variation in the procedure, differences in persistence or self-control.

In conclusion, we found that 3 to 5-year old children, adult humans and tool-making New Caledonian crows were able to make profitable decisions requiring tool-use and the ability to delay gratification, make effective decisions and judge work-effort, by adapting and extending a paradigm used previously in non-tool making Goffin's cockatoos [50] and tool-making orangutans [5]. Our findings suggest that: (1) crows and cockatoos show self-control and can perform similarly to adult humans and older children on a range of tool related delayed gratification tasks, but struggle when they have to attend to multiple cues thereby increasing task complexity; (2) children and orangutans may have a bias to use tools that other tool-making species may not have. Future work extending these findings should offer valuable insight into how the cognition behind self-control in corvids, parrots and primates evolves.

## Supporting information

**S1 Table. Number of correct trials for all constellations of the tests for the crows for each individual.** In the tool selection and apparatus functionality condition the total number of trials (in brackets) varied between individuals, depending on when they reached the criterion. P-values calculated from exact binomial tests. Significant p-values highlighted in bolt.
MPR = most preferred reward, Sessions 1 to 2 in the tool selection and tool selection quality allocation condition are for comparability with the Goffin's cockatoos; * p < 0.05, ** p <0.01, *** p < 0.001.
(DOCX)

**S2 Table. Crow subject information.**
(DOCX)

**S3 Table. Generalized linear mixed models on factors affecting the number of correct trials in crows.** N = 6. Significant p-values are highlighted in bold.
(DOCX)

**S4 Table. Posthoc comparison of conditions of crow data with Tukey correction for multiple comparison.**
(DOCX)

**S5 Table. Generalized linear mixed models on factors affecting the number of correct trials in children aged 3–5 years, with age in years.** N = 88. Significant p-values are highlighted in bold.
(DOCX)

**S6 Table. Posthoc comparison of conditions of children data with Tukey correction for multiple comparison.**
(DOCX)

**S7 Table. Performance across all conditions for the crows with each apparatus singly.**
Results reflect results of Wilcoxon 1-sample signed ranks tests–chance value = 50%. Significant p-values (<0.05) highlighted in bold.
(DOCX)

**S8 Table. Comparison of performance within conditions between crows and cockatoos.**
Results reflect Mann Whitney U-tests. Significant p-values highlighted in bold.
(DOCX)

**S1 Video.**
(MP4)

**S2 Video.**
(MP4)

## Acknowledgments

We would like to thank the staff, parents and children at Sutton V.A. Lower School, St Andrew's C of E Primary School, Under Fives Roundabout, Histon Early Years Centre and Patacake Day Nursery for their participation in this study. Thank you to Ian Millar for help in apparatus construction and to Alizée Vernouillet for video coding. Thanks to Province Sud for the permission to work in New Caledonia, and to Dean M. and Boris C. for granting property access for catching and releasing the crows.

## Author Contributions

**Conceptualization:** Rachael Miller, Romana Gruber, Russell D. Gray, Markus Boeckle, Alex H. Taylor, Nicola S. Clayton.

**Data curation:** Rachael Miller, Romana Gruber, Anna Frohnwieser, Martina Schiestl.

**Formal analysis:** Rachael Miller.

**Funding acquisition:** Russell D. Gray, Alex H. Taylor, Nicola S. Clayton.

**Investigation:** Rachael Miller, Romana Gruber, Anna Frohnwieser, Martina Schiestl.

**Methodology:** Rachael Miller, Romana Gruber, Anna Frohnwieser, Russell D. Gray, Markus Boeckle, Alex H. Taylor, Nicola S. Clayton.

**Project administration:** Rachael Miller, Romana Gruber, Anna Frohnwieser, Sarah A. Jelbert, Nicola S. Clayton.

**Resources:** Rachael Miller, Nicola S. Clayton.

**Supervision:** Rachael Miller, Alex H. Taylor, Nicola S. Clayton.

**Writing – original draft:** Rachael Miller, Romana Gruber.

**Writing – review & editing:** Rachael Miller, Romana Gruber, Anna Frohnwieser, Martina Schiestl, Sarah A. Jelbert, Russell D. Gray, Markus Boeckle, Alex H. Taylor, Nicola S. Clayton.

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
