## [Decision Letter · Decision Letter 0]

3 Sep 2019

PONE-D-19-18492

Decision-making flexibility in New Caledonian crows, young children and adult humans in a multi-dimensional tool-use task

PLOS ONE

Dear Dr Frohnwieser,

Thank you for submitting your manuscript to PLOS ONE. After careful consideration, we feel that it has merit but does not fully meet PLOS ONE’s publication criteria as it currently stands. Therefore, we invite you to submit a revised version of the manuscript that addresses the points raised during the review process.

As you will see both reviewers are mainly positive but have several comments which I would ask you to attend to. Here I would like to highlight Reviewer 1's comment on the species comparison being potentially weakened by the fact that species received a different number of trials. Rev 1 also has several statistical comment I would like yo to attend to and Rev 2 has several comment which you will find helpful when revising your introduction and discussion. 

We would appreciate receiving your revised manuscript by Oct 18 2019 11:59PM. To enhance the reproducibility of your results, we recommend that if applicable you deposit your laboratory protocols in protocols.io, where a protocol can be assigned its own identifier (DOI) such that it can be cited independently in the future. For instructions see: http://journals.plos.org/plosone/s/submission-guidelines#loc-laboratory-protocols

We look forward to receiving your revised manuscript.

Kind regards,

Juliane Kaminski

Academic Editor

PLOS ONE

Journal Requirements:

Reviewers' comments:

Reviewer's Responses to Questions

**Comments to the Author**

1. Is the manuscript technically sound, and do the data support the conclusions?

Reviewer #1: Partly

Reviewer #2: Yes

2. Has the statistical analysis been performed appropriately and rigorously? 

Reviewer #1: No

Reviewer #2: Yes

3. Have the authors made all data underlying the findings in their manuscript fully available?

Reviewer #1: Yes

Reviewer #2: Yes

4. Is the manuscript presented in an intelligible fashion and written in standard English?

Reviewer #1: Yes

Reviewer #2: Yes

5. Review Comments to the Author

Reviewer #1: In this article, the authors examined flexible decision-making in the context of tool-use tasks in New Caledonian crows and human children and adults. Comparisons with previous studies with orangutans and cockatoos are made. While the results are interesting, I do not think the results justify the conclusions drawn with regard to the species comparisons. Additionally, there are a number of issues with the statistical analyses (as detailed below).

There are differences in the procedure (e.g. the trial number) and the statistical analysis (which seems to be a consequence of the difference in the procedure) which critically undermine the comparison between crows and humans. The crows had more opportunities to learn across trials compared to humans. At various locations throughout the manuscript, birds are compared to humans (e.g., lines 38, 612 and 616) without acknowledging the differences in procedure (and as a consequence the possibility that similar performance levels might reflect different cognitive underpinnings).

Line 347: Interobserver reliability: Kappa and the number of observations and more details about the reliability coding (did the same person do the live coding and the reliability coding? was the reliability coder naïve with respect to the research questions?) should be added.

Line 350-364: were random slopes included in the GLMMs (i.e. random slopes of condition, apparatus type, and trial number)? There is evidence that without a maximal random slopes structure GLMMs can be overconfident (Barr , Levy, Scheepers, Tily 2013 Random effects structure for confirmatory hypothesis testing: Keep it maximal. J Mem Lang 68(3), 255-278.)

Line 363: which variables were specified as “control variables”?

Line 378: The reported chi-squared value “-0.32” does not seem to be correct (if the p value is <0.001). Looking at table S3 shows that “-0.32” is probably the estimate.

Line 378: was condition included as a factor (i.e., as a dummy-coded categorical variable)? Given that there is only one estimate for condition (table S3), condition seems to be entered as a covariate, which would be difficult to justify. Could the authors please specify how condition was entered in the model? Assuming that it is treated as a factor why are no pairwise comparisons reported compared to a reference category (e.g. in Table S3)?

Line 426: the main effect of age has a limited meaning in the presence of the interaction term (it refers to the age effect with condition at its reference category). For that reason, I would recommend to just report the interaction term here. Besides, how did the authors calculate a likelihood ratio test (LRT) for age (in the presence of an interaction term)?

Line 426: typo “=,”

Line 429-430: a difference between the conditions is reported without any supporting inferential statistics (e.g. post-hoc tests).

Table 2, table S7 and Lines 486, 491: are multiple observations per individual included here in these binomial tests? Given that multiple trials per condition and subject were conducted, it seems to be the case (looking at the data file supports this interpretation). This is a case of pseudo-replication. Condition and apparatus type need to be analyzed separately to avoid pseudo-replication.

Table 3 seems to be redundant and just a dichotomized version of Figure 3.

Discussion section: Even though the authors acknowledge the differences in procedure between the different species, they draw comparisons here between children and the birds. However, while the birds received 24 trials in the motivation condition, for example, the children received 2 trials. Given that we do not know how children’s performance would look like after 24 trials I am not sure what can be said about the differences in performance between humans and crows. For example, I am not sure whether children would continue to choose the tool in motivation trials after 24 trials (and/or when children would receive a food reward).

Supplementary videos: it looks like the experimenter touched both tools and looked at the child when children made their choice. Where any precautions taken to guard against the possibility of inadvertent cueing?

Reviewer #2: This is a very interesting and methodologically sound study. However, it has remained unclear what the focus of the research is (self-control or a broader focus also on undertsanding of tool functionality), and this has to be made clearer in the introduction as well as discussion. The results can in parts be presented more concisely. I had some comments on the analysis. My comments can be found in the attached file.

6. PLOS authors have the option to publish the peer review history of their article (what does this mean?). If published, this will include your full peer review and any attached files.

Reviewer #1: No

Reviewer #2: No

---

## [Author Response · Author response to Decision Letter 0]

17 Jan 2020

Response to Reviewers

Dear Dr Kaminski, 

Following an invitation to revise and re-submit (PONE-D-19-18492), we would like to re-submit our manuscript entitled ‘Decision-making flexibility in New Caledonian crows, young children and adult humans in a multi-dimensional tool-use task’. 

We wish to thank you and the two reviewers for the helpful and constructive comments. We have now fully revised the manuscript and accompanying documents in accordance with these comments. Please find responses to each comment in this response to reviewers document – please note that line numbers correspond with the tracked changes version of the manuscript. 

We hope that following our revisions, you will consider our manuscript for publication in PLOS ONE. 

Yours Sincerely, 

Rachael Miller, Romana Gruber, Anna Frohnwieser, Martina Schiestl, Sarah A. Jelbert, Russell D. Gray, Markus Boeckle, Alex H. Taylor, Nicola S. Clayton

PONE-D-19-18492

Decision-making flexibility in New Caledonian crows, young children and adult humans in a multi-dimensional tool-use task

PLOS ONE

Dear Dr Frohnwieser,

Thank you for submitting your manuscript to PLOS ONE. After careful consideration, we feel that it has merit but does not fully meet PLOS ONE’s publication criteria as it currently stands. Therefore, we invite you to submit a revised version of the manuscript that addresses the points raised during the review process.

As you will see both reviewers are mainly positive but have several comments which I would ask you to attend to. Here I would like to highlight Reviewer 1's comment on the species comparison being potentially weakened by the fact that species received a different number of trials. Rev 1 also has several statistical comment I would like yo to attend to and Rev 2 has several comment which you will find helpful when revising your introduction and discussion. 

We would appreciate receiving your revised manuscript by Oct 18 2019 11:59PM. To enhance the reproducibility of your results, we recommend that if applicable you deposit your laboratory protocols in protocols.io, where a protocol can be assigned its own identifier (DOI) such that it can be cited independently in the future. For instructions see: http://journals.plos.org/plosone/s/submission-guidelines#loc-laboratory-protocols

• A rebuttal letter that responds to each point raised by the academic editor and reviewer(s). This letter should be uploaded as separate file and labeled 'Response to Reviewers'.

• A marked-up copy of your manuscript that highlights changes made to the original version. This file should be uploaded as separate file and labeled 'Revised Manuscript with Track Changes'.

• An unmarked version of your revised paper without tracked changes. This file should be uploaded as separate file and labeled 'Manuscript'.

We look forward to receiving your revised manuscript.

Kind regards,

Juliane Kaminski

Academic Editor

PLOS ONE

Journal Requirements:

Reviewers' comments:

Reviewer's Responses to Questions

Comments to the Author

1. Is the manuscript technically sound, and do the data support the conclusions?

Reviewer #1: Partly

Reviewer #2: Yes

2. Has the statistical analysis been performed appropriately and rigorously? 

Reviewer #1: No

Reviewer #2: Yes

3. Have the authors made all data underlying the findings in their manuscript fully available?

Reviewer #1: Yes

Reviewer #2: Yes

4. Is the manuscript presented in an intelligible fashion and written in standard English?

Reviewer #1: Yes

Reviewer #2: Yes

5. Review Comments to the Author

Reviewer #1: In this article, the authors examined flexible decision-making in the context of tool-use tasks in New Caledonian crows and human children and adults. Comparisons with previous studies with orangutans and cockatoos are made. While the results are interesting, I do not think the results justify the conclusions drawn with regard to the species comparisons. 

We now present the results of the two studies as separate studies and do not directly compare the data. We now draw conclusions that are based on our data.

Additionally, there are a number of issues with the statistical analyses (as detailed below).

We changed the statistical analysis according to the reviewers comments (see below)

There are differences in the procedure (e.g. the trial number) and the statistical analysis (which seems to be a consequence of the difference in the procedure) which critically undermine the comparison between crows and humans. The crows had more opportunities to learn across trials compared to humans. At various locations throughout the manuscript, birds are compared to humans (e.g., lines 38, 612 and 616) without acknowledging the differences in procedure (and as a consequence the possibility that similar performance levels might reflect different cognitive underpinnings).

A paragraph was added at the beginning of the materials and methods section (line 142-146) and the Discussion section (line 530-532) to acknowledge the differences in methodology between the species.

Line 347: Interobserver reliability: Kappa and the number of observations and more details about the reliability coding (did the same person do the live coding and the reliability coding? was the reliability coder naïve with respect to the research questions?) should be added.

Cohen’s Kappa and number of observations has been added (line 370-375). The live coding and video coding were conducted by different observers; however, due to the nature of the experiment (i.e. being able to tell from the video whether a trial was solved correctly or not) the second observer was not completely naïve to the research question.

Line 350-364: were random slopes included in the GLMMs (i.e. random slopes of condition, apparatus type, and trial number)? There is evidence that without a maximal random slopes structure GLMMs can be overconfident (Barr , Levy, Scheepers, Tily 2013 Random effects structure for confirmatory hypothesis testing: Keep it maximal. J Mem Lang 68(3), 255-278.)

We now include random slopes and intercept by using (1+Trial | Individual) as random factor. Based on the problem of non-convergence, we had to reduce the complexity of the model and drop the factor apparatus as fixed factor and condition in the random slopes. In case of non-convergence we dropped each single factor separately until convergence was reached while keeping maximum complexity of the model, as the reviewer 2 suggested.

Line 363: which variables were specified as “control variables”?

We don’t use the term control variables anymore, only the terms predictor variables and random effects

Line 378: The reported chi-squared value “-0.32” does not seem to be correct (if the p value is <0.001). Looking at table S3 shows that “-0.32” is probably the estimate.

Correct, we changed this now to present the z-value.

Line 378: was condition included as a factor (i.e., as a dummy-coded categorical variable)? Given that there is only one estimate for condition (table S3), condition seems to be entered as a covariate, which would be difficult to justify. Could the authors please specify how condition was entered in the model? Assuming that it is treated as a factor why are no pairwise comparisons reported compared to a reference category (e.g. in Table S3)?

It was entered as a factor. We now report the values for all categories and the initial results of the model as well as the pairwise comparison with Tukey post-hoc test in the supplements.

Line 426: the main effect of age has a limited meaning in the presence of the interaction term (it refers to the age effect with condition at its reference category). For that reason, I would recommend to just report the interaction term here. Besides, how did the authors calculate a likelihood ratio test (LRT) for age (in the presence of an interaction term)?

We now report the interaction term only. We used comparisons of deviance information criteria and report this in the analysis section now.

Line 426: typo “=,”

This has been corrected

Line 429-430: a difference between the conditions is reported without any supporting inferential statistics (e.g. post-hoc tests).

We now report the inferential statistics on the data.

Table 2, table S7 and Lines 486, 491: are multiple observations per individual included here in these binomial tests? Given that multiple trials per condition and subject were conducted, it seems to be the case (looking at the data file supports this interpretation). This is a case of pseudo-replication. Condition and apparatus type need to be analyzed separately to avoid pseudo-replication.

This analysis was rerun with condition and apparatus type separately to avoid pseudo-replication.

Table 3 seems to be redundant and just a dichotomized version of Figure 3.

Table 3 has been removed from the manuscript.

Discussion section: Even though the authors acknowledge the differences in procedure between the different species, they draw comparisons here between children and the birds. However, while the birds received 24 trials in the motivation condition, for example, the children received 2 trials. Given that we do not know how children’s performance would look like after 24 trials I am not sure what can be said about the differences in performance between humans and crows. For example, I am not sure whether children would continue to choose the tool in motivation trials after 24 trials (and/or when children would receive a food reward).

We added a sentence to acknowledge this difference in procedure (line 558).

Supplementary videos: it looks like the experimenter touched both tools and looked at the child when children made their choice. Where any precautions taken to guard against the possibility of inadvertent cueing?

Great care was taken to ensure the two tools/stickers were placed on the table simultaneously. In trials where the choice had to be explained further (“you can have the immediately available item now or the tool to try to use later”), the order in which the two items were mentioned was randomised between trials, so that each option was mentioned first/last in multiple trials. We made this clearer in the methods section (line 327-328).

Reviewer #2: This is a very interesting and methodologically sound study. However, it has remained unclear what the focus of the research is (self-control or a broader focus also on undertsanding of tool functionality), and this has to be made clearer in the introduction as well as discussion. The results can in parts be presented more concisely. I had some comments on the analysis. My comments can be found in the attached file.

6. PLOS authors have the option to publish the peer review history of their article (what does this mean?). If published, this will include your full peer review and any attached files.

Do you want your identity to be public for this peer review? For information about this choice, including consent withdrawal, please see our Privacy Policy.

Reviewer #1: No

Reviewer #2: No

Review for

PLOS ONE

Decision-making flexibility in New Caledonian crows, young children and adult humans

in a multi-dimensional tool-use task

This study set out to investigate the ability of human adults, children, and New Caledonian crows to make profitable decisions in a variety of tool-use contexts. In addition, the authors compare their results with the findings from a previous study on cockatoos and orangutans which used a similar metholodogy. Therefore, one of the additions that the current papers brings to the literature is the use of a comparable methodology of a variety of species, including tool-using and non-tool-using ones.

The authors predicted that the crows would show a similar performance as the performance in the orangutans and cockatoos, but had no directed hypothesis about the performance of the human children. The key message that the authors draw from their results is that crows, human adults and children from 4 years of age can make profitable decisions in their studied tool-use context; however, crows show a decreased performance when they had to attend to several cues in parallel and 3-year-old children showed chance performance in almost all conditions. Lastly, the authors found a strong bias for selecting a tool, even if this choice was less efficient, across children of all ages. 

Thank you for your comment. We have added a directed hypothesis concerning the children’s performance (line 136-137).

The article is appropriately structured and clearly presented. However, I have one small comment on one of the subheadings in the result section (“Performance in New Caledonian crows, Goffin’s cockatoos, orangutans, children and adult humans”) which is formulated a bit too vague (see below). The paper title does reflect the content and attracts attention. All in all, the article fits with the scope of PlosOne.

Thank you for your comment. The subheading has been changed to be clearer (line 497).

Please find my detailed comments below.

Introduction

 The introduction lacks a little bit structure. Several concepts are introduced within the first page (effective decision making, self-control, delay of gratification, tool use, delay of gratification in tool use), which makes it hard to figure out what the primary focus of the current paper is. The overall area of interest is effective decision-making and it is stated that one aspect underlying this ability is self-control. The authors also mention other aspects such as work-effort sensitivity and attention to the functionality of the tools. Even though the latter two concepts are mentioned further below in the paper as well, the introduction suggests that the main focus of the authors is on self-control/delay of gratification. However, as the methods section shows, this is not the case as only some of the studied conditions are explicitly looking at delay of gratification. Others look at the perception of tool functionality and work-effort sensitivity. Therefore, the introduction could be rephrased in stating that several aspects of decision-making in a tool-use context are being investigated. Otherwise, if the focus of the introduction remains heavily on self-control, the choice of that many conditions (some of which do not involve delay of gratification) remains unjustified. This will imply that the authors give some more information on what we already know about work-effort sensitivity and attention to the functionality of tools in the studied species (especially crows and human children).

What I would also find interesting is some statement on how these different concepts of decision-making (self-control, work-effort sensitivity, sensitivity to tool functionality, etc) relate to each other.

What is also lacking at the end of the introduction or at the beginning of the methods section is an account of how the researchers are aiming to measure delay of gratification, work-effort sensitivity, and the sensitivity to the functionality of the tool. That is, here the different conditions/comparisons between conditions can be introduced (in the current paper, the conditions are just described in the methods section, but is remains somewhat implicit what the conditions are designed to measure). 

These comments have been addressed throughout the introduction and more detail was added on the different measures that were used in the study.

More specific comments I had are:

Lines 58/59: “There is significant improvement by 4 years old and above (14), with some studies showing improvement between ages 3 and 5.” This statement is a bit unclear as it doesn’t become clear how the improvement looks like. 

In addition, “There is significant improvement by 4 years “seems to describe almost the same as “improvement between 3 and 5”, and this redundancy is even more pronounced by the fact that the sentence before this one already states that self-control being present in toddlerhood and preschool age. Thus, I suggest to rephrase this sentence to state clearly which improvement the cited papers have shown and how the developmental trajectory looks like. Also note that there is not only evidence for an increase in self-control between 3 and 5, but also even beyond 5 years of age (see e.g., Jacqui A. Macdonald, Miriam H. Beauchamp, Judith A. Crigan & Peter J. Anderson (2014) Age-related differences in inhibitory control in the early school years, Child Neuropsychology: A Journal on Normal and Abnormal Development in Childhood and Adolescence, 20:5, 509-526).

We have clarified this sentence (line 61).

Lines 71/72: I am not entirely convinced by the authors’ conclusion that “no clear pattern has emerged between the self-control abilities of tool-using and non-tool-using species”. While it is true that no conclusion can yet be drawn from studies looking into self control in non-tool-using contexts (because – as the authors rightly state – species have rarely been tested on the same tasks), the studies on animals’ performance in tasks involving delayed gratification in a tool-use context (lines 86-100), might be somewhat clearer. It seems to emerge that animals of different species are able to delay gratification and to select a tool over getting an immediate reward regardless of whether the animal belongs to a tool-using species or not. The authors should discuss the possibility that the existing literature might suggest that there is no relationship between self-control abilities and being a tool user (or not) in delay of gratification tasks involving tool use. Alternatively, if the authors think the results of the previous studies are truly inconclusive, they should explain more clearly why they think that (if there is a reason other than that different tasks were used).

This sentence has been clarified and some information added (line 74-76).

A related point, what is missing before the paragraph starting in line 68 is a bit of theoretical background, i.e., whether researchers have had a priori expectations to find that tool-users have better self control (and if so, why they expect that).

We now included the background theory as to tool-users may have better self-control abilities (lines 66-70).

Also related to this part in the text, I think it would be great to move the point stated in line 101 (that “very few studies have used comparable methodology”) further up, to line 71, i.e., the authors could say that no clear pattern has emerged yet, also because different methodology has been used. This is an important point and can be made way earlier in the text. 

We now stated the use of different methodologies earlier in the manuscript (lines 74-76).

Lines 105/106: “found differences between human and non-human species’ performances, which suggest that the task may not measure the ability that it is designed to test”. This sentence is a bit too vague – could you please specify what kind of differences were found? After all, you seem to describe not mere performance differences, but actually differences in the how the task is approached/perceived by the subjects?

Relatedly, line 106, “may not measure the ability that it is designed to test”. While possible, this is not always true. Isn’t it more likely that when task differences occur (see the performance differences in the trap tube task which the authors mention) that the task is indeed measuring what it is supposed to measure in one species, but not the in the other one? It seems that in most of the cases, the issue is that a task has been designed to measure an ability in one species and then fails to transfer rather than it not measuring the desired ability in either of the species. Thus, I suggest to rephrase the sentence to reflect this possibility.

This has been rephrased and clarified (line 110-119).

At some point in the introduction it might be nice to have a brief explanation of why we should know about decision-making abilities in a tool-use context specifically. While the authors describe that a tool-use context is special (by having more relational complexity and more work-effort), they miss to describe why it is interesting to investigate delay of gratification in a tool-use context. What insight can this give us on different species’ cognition? Is it because of the greater ecological validity? One could argue that if one is interested in self-control only, one should revert to non-tool-use tasks as these would not introduce another potential source of variance. I would like to see the authors countering this potential criticism.

A justification for this has been added to the introduction (line 66-76)

Lines 125/126: More information should be given on the different onset times of delayed gratification and tool functionality - the link to what has previously been investigated is rather short. Some more explanation here might also help to justify the choice of the age range, which is missing in the methods section (see below).

We added a justification for the chosen age range in the methods section (line 174-176).

Lines 126-128: Unlike for the non-human species, the authors state no directed hypotheses about children’s performance in the task. The authors should explain why not. Can one deduce from the previous literature on the different onset times of tool functionality understanding on the one hand and delay of gratification on the other hand help to formulate a hypothesis? 

We have added a directed hypothesis for the children’s performance (line 136-137).

Methods

The methodology is sound. The statistical design seems mostly appropriate, however I have a few questions/comments on the design and analysis (see below). The procedures have mostly been described sufficiently, however, some details are missing that allow the study to be replicated (see below).

Line 110: Please provide a justification for why this age range of children (3-5) was chosen.

This was added in line 174.

Line 135: how was the sample size for the crows determined? Did the authors have a target number?

As the crows were wild caught, the number of individuals available for the study depended on how many crows we had for the season (always between 8 and 10 individuals) and which ones were reliable to work with and succeeded in previous training stages. This lowered the number to six reliable working crows. This is now stated in the text (line 155-157).

Line 139: The authors state that they were testing juvenile and adult crows. What is the implication of this? Is it assumed that juveniles already have the same abilities as adults? Why did the authors not decide to only test adult crows? Does a mix of juveniles and adults not matter? How much were practical issues decisive? More information on these questions should be given in order to help the reader understand whether having juveniles and adults in the sample is not an issue at all or should be seen as a concern.

We agree that ideally this study should have used enough adult and juvenile crows to make meaningful comparisons between the two age groups. However, as mentioned above, the crows were wild caught and only few animals were available, thus making it impossible to reduce the sample size further by differentiating by age. We have added more information in the methods section (line 154-157).

Line 142: can the authors add information on how big the caught family groups were?

We now stated the sizes of the groups “The birds were housed in small groups, consisting of two to four individuals per group, in a ten-compartment outdoor aviary, with approximately 7x4x4m per compartment, containing a range of natural enrichment materials, like logs, branches, and pine cones” (lines 157-160).

Line 156: how was the sample size for the child sample determined? Was there an a-priori sample size calculation?

We estimated that children will roughly show a percentage of 80 percent correct across our three age groups of 3, 4, and 5-year olds. Based on this we calculated with the g*power software for binomial tests that we would need 28 kids per age group. Effect size g= 0.3; �=0.05; estimated sample size = 28 for one-tailed binomial test per age group.

Line 160: how was the sample size for the adult sample determined?

We estimated that adults will roughly show a percentage of 90 percent correct. Based on this, we calculated with the g*power software for binomial tests that we would need 13 adults, which we overpowered by recruiting 20 adults. Effect size g= 0.4; �=0.05; estimated sample size = 13 for one-tailed binomial test.

Line 171: would the authors consider moving figure S1 to the main manuscript? It would benefit the understanding of the experiment tremendously to have a picture of the apparatuses readily available. If not, I would like to ask the authors to move S1 to the beginning of their supplementary materials, before the result tables (just to stay in chronological order).

The figure S1 has now been moved to the main manuscript as Figure 1 (line 205).

Lines 173-181: the information that two different stick apparatuses were used for humans and NC crows should come before the description of the apparatuses; just makes this section easier to understand.

We now moved the information as to why two different stick apparatuses were used before the description of the apparatuses, “We used two different ‘stick-apparatus’ for the humans and the crows, both apparatuses were functionally the same, as they required a stick to contact a reward and move it to the left or right. The minor variation in the apparatus structure was due to the testing equipment available in New Caledonia. The ‘stick-apparatus’ for the humans…” (lines 191-194)

Line 179: again, it would be nice if figure S1 was in the main manuscript rather than in the supplementary material. 

Figure S1 is now in the main manuscript as Figure 1 (line 205).

Line 179: as the authors refer to S1a Fig, they should also refer to S1b Fig (e.g., in line 175).

We are now mentioning Figure 1b (line 199).

lines 335/336: please add the explanation for why subjects were divided into two subsets with a different order of conditions.

The reason for different order of conditions has now been added (lines 359-367)

347: please explain why an unusually low number of videos (10%) were coded for reliability – usually 20 – 25% are used?

10% is the usual number of videos coded for reliability in our field. We added information on how many individual trials were double coded and Kappa’s Cohen (line 370-375).

355-360: Why were no random slopes (for condition, apparatus, trial number) included into the model? According to Barr, Levy, Scheepers and Tily (2013) one should try to construct a maximal full model in order to keep type 1 error rate minimal. I suggest re-running the models and – if they lead to non-convergence – simplifying the models according to a pre-determined process.

References: Barr, D. J., Levy, R., Scheepers, C., & Tily, H. J. (2013). Random effects structure for confirmatory hypothesis testing: Keep it maximal. Journal of Memory and Language, 68(3), 225-278.

We now include random slopes but had to exclude apparatus from the model for convergence reasons. Also see comments to Reviewer 1.

There is no report on model stability nor any model diagnostics checks. These analyses have to be run and included in the main manuscript or the supplementary material.

We were not able to find ways of calculating model stability and model diagnostics checks for binomial data with random effects. In case the reviewer has a suggestion how to do so, we are happy to proceed with it in the suggested way

Line 358/359: There are two issues with age.

1. Usually in developmental studies on young children, when age is entered as continuous variable into models, this means that age in months is used. If children are assigned a year value (3, 4, 5) this is usually referred to as using age as a categorical variable. As I understand, in the current manuscript children were assigned a year value, which is equivalent to using a categorical variable. Why did the authors choose to label this a continuous variable? The variable with the three levels is arguable too coarse to label it continuous?

2. Why in the first place did the authors decide to bin age into the three levels? I suggest using the truly continuous variable (age in years and months) instead. Why binning one of the few continuous variables that developmental psychologists have and thus losing information? In addition, binning age implies that a child aged 3y11m is different from a child aged 4y1m, whereas a child aged 4y1m is put into the same category as a child with 4y11m. This seems unintuitive. It is suggested to rerun the analyses with age as a continuous variable.

We now used year as a decimal as we calculated it according to the exact age of the child. We therefore decided to exclude the information completely. Due to changes in the analysis Table S5 and S6 have been removed from the supplementary materials. Additionally, the analysis does not merit any additional insight into the performance, as the reviewer noted. We therefore decided to exclude the information completely. 

Results

The results were partly not easy to understand. Some information in the text is overlapping with information given in table 1. It remained unclear why some results were presented in the main text and others were moved to the supplementary section. Table 2 is also rather difficult to grasp (see below). 

We have clarified the results section and tables by removing overlapping information and changing the sub-headings.

Lines 430/431: relating to the comment above on the measurement level on age, it is unclear why the authors present the results on age for both age as a truly continuous (age in months) and a categorical (age in years) variable. The authors should consider using only the continuous variable from the start or to clearly justify their decision in the methods section.

We now only include the age as a continuous variable and excluded the rest from the MS and the Supplements.

Lines 431-432: Do the authors have any references in favour of excluding subjects in order to create more distinct age classes? To me, it seems that there is no reason to get rid of datapoints (especially given that the sample is not very large to begin with). In order to circumvent excluding datapoints, one could just go with age as a continuous variable for all analyses.

Thank you for the comment, we now only do this and do not exclude any datapoints.

Line 363: it is unclear which variables in the GLMM are the control variables. I assume that one is gender, but it’s unclear whether for the children age was treated as a control variable or predictor. Please specify.

We don’t use control variables but only predictor variables and random factors.

Line 364: the reference given after this sentence seems to be wrong or at least it is unclear how it relates. The reference “52” given below in the reference list is “Göckeritz S, Schmidt MF, Tomasello M. Young children's creation and transmission of social norms. Cognitive Development. 2014;30:81-95.” Please correct and check the correctness of the references throughout the text.

We now cite the correct paper, which is reference number 59.

Lines 365-367: It is not explained what was analysed using the two-tailed statistics; which comparisons were made? Please clarify.

We now clarify in the data analysis section that we calculated success rate for each condition and age class 3-5 years.

Line 367: Please explain why for non-parametric tests a different software was used.

We ran only the Wilcoxon test in SPSS because of applicability and data structure.

Line 369: could the authors clarify which question the exact two-tailed Binomial tests address? “assessing success rates” is quite vague and could also be achieved by studying the descriptive data.

We calculated whether children were performing above chance within each age class and condition.

Lines 382-387 and following (e.g. lines 411-412)/Table 1: Would the authors consider adding to Table 1 the performance for each apparatus separately? In the text, the data for both “across apparatuses” and “for each apparatus separately” are presented together, and in between there are references to the table. It was a bit odd that the table would only present part of the data. 

We moved references to Table 1 in the text so they are less confusing, but have kept Table 1 as it was to give an overview of the most important data (performance in each condition).

Lines 382-407 seem to describe in large parts what is displayed in table 1 and Figure 2. This is repetitive and the authors should consider presenting these results in a more concise manner, e.g., by cutting some parts of the text. In any case, I would recommend removing those sentences explaining which was the correct choice in which condition. This was already explained in the methods section and in Figure 1.

We have made this section less redundant by removing parts of the text.

Line 426: since the interaction between age and condition is significant, the main effect of age is not interpretable (as it is dependent on condition) and should thus not be presented. Instead, only the interaction effect should be reported. 

We now only present the interaction term.

Line 428: “success rate increases with age” – this sentence implies a main effect of age, but this is not what was found, Please rephrase so that the text correctly describes the relationship between age and condition. For this, displaying the results visually would be beneficial for the understanding of the reader.

We are now more specific about the result.

Line 430: Why was the interaction not further investigated? The methods section mentioned the posthoc tests using the multcomp package, but the results from these analyses don’t seem to appear in the results section. Please add.

We added the mulcomp results in the supplement now.

Lines 433-434. “Across all conditions, only the 4 and 5-year old children selected correctly above chance across all trials”; later on the authors explain “3 to 5-year olds did not select correctly above chance in the tool functionality condition.” These two sentences are contradicting and the first sentence is potentially misleading. It suggests that 5-year-olds performed above chance in all conditions, while a look into the table and the second sentence exclude the tool functionality condition. To avoid a misunderstanding and to highlight the fact that the results are not as clear-cut as the first sentence might suggest, this first sentence should be modified accordingly to accurately describe the data.

This sentence was reworded, as it referred to the analysis of all conditions combined, not individual analyses of each condition (line 459).

Table S7: would the authors consider including the data displayed in this table into Table 2 in the main text? This way, the performance in all conditions could be more easily compared by the reader.

Table S7 was moved to the manuscript and is now Table 3.

Table 2: Instead of displaying the uncorrected p-values and then adding “NS” if the values got insignificant after the Bonferroni correction, would the authors consider just presenting the p-values obtained after the correction? This would make it a lot easier to quickly grasp the results pattern when looking at the table. What benefit does including the non-corrected values have?

Thank you for the suggestion but we prefer to leave the information inside to give the reader the possibility of fully understand the data.

Lines 467-477 and Table 3: Why is children’s performance in the tool functionality condition displayed collapsed across the two subconditions? It was found that children indeed chose correctly when the tool was functional, so the conclusion that children performed poorly in the tool functionality condition might not accurately describe their ability to perceive affordances and use tools. 

Due to suggestions by Reviewer 1 Table 3 was removed from the manuscript. We discussed children’s performance in the tool functionality test in the discussion section. While children picked the functional tool when the choice was between the tool and a low quality reward in the “functional” sub-condition, they also picked the tool when it was non-functional, indicating that they did not understand or take into account tool functionality when making their choice, but made it based on reward quality alone.

Discussion and conclusions

Line 497: it is stated that the study showed that the studied species were able to make profitable decisions in tasks requiring delayed gratification in a tool-use context. This phrasing suggests that all studied conditions involved a delay of gratification component, which does not seem to be the case. The authors should carefully rephrase this sentence at the beginning of this study; this is an important sentence which many readers will read first after the abstract. As it is formulated at the moment, the sentence suggests that the study was a delay of gratification /self-control study, which I wouldn’t have called it. The subsequent sentence indeed states that the studied populations “were able to flexibly select between reward items of differing quality and tools […] relative to the context of each condition.” – this has nothing to do with delay of gratification and shows that the focus of the paper was broader (effective decision-making). This is also evident by how the following sections in the discussion are structured: First the results on tool functionality are presented, then on motivation, and only then the delay of gratification part is presented. The authors need to be clearer in describing whether self-control/delay of gratification is their main focus of the paper or just one of several (and if there are several aspects, are the other ones related to the requirements of effective decision-making or as these aspects on the motivations and perceptions of affordances in tool-use contexts?). This is unclear in both the discussion and the introduction and needs to be more streamlined.

We have made changes to both the introduction and discussion in order to make the focus of this study clearer. Specifically, we changed the first paragraph of the introduction to include work-effort sensitivity and effective decision-making (line 52-53).

Linea 514/515: the authors state that children failed in the tool functionality condition. However, if I understood the results section correctly, the results could be differentiated between the two sub-conditions (i.e., when the tool was functional vs not functional). This is an interesting finding and should be acknowledged in the discussion. The way this is currently formulated does not represent the data accurately (very broad brush).

We have added additional information and discussion about the tool functionality condition (line 619-622).

Lines 522-527: Can the authors elaborate on why the “birds’ performance at the four tasks that the 3-year olds struggled with, and particularly the motivation condition, which 4- 5-year olds and orangutans also struggled with” suggest that the cognitive demands of these conditions were “non-trivial”? Could these results also just be explained by a strong bias of choosing the tool in the children, which would not comment on whether or not the cognitive demands are difficult (i.e., it is about motivation, not cognition)?

This has been changed.

Lines 533-534: “4-5-year olds, though not 3-year olds, appear able to overcome the desire to immediately select the tool”. This might suggest that 3-year-olds lack the ability to overcome this desire. However, the authors should consider that the reason why the 3-year-olds struggled could eb that they perceived the task differently, for example as a task in which they try to explore the novel materials and tools instead of one in which getting the reward was the primary goal.

This has been added (line 568-571).

Line 555: another reason why children performed badly at the tool functionality task could not only be the “drive to give it a go”, trying to make the non-functional tool work, but it could also be that the children know well in advance that the tool will not work, but select it because it is fun manipulating the tool and exploring the apparatus with it. Could the authors provide information on what the children were attempting to do with the tool (did they just select it, were they trying to actually get the reward?) or were children not given the chance to try out the non-functional tool in the test?

This has been added (line 573-577).

In the result section it was found that performance sometimes differed between the two apparatuses. This should be at least mentioned in the discussion, preferably also discussed.

This has been added (line 647-651).

The authors provide possible explanations of why there yould be a difference between crows and cockatoo performance, but an explanation of what the difference between in the conditions in which the 3-year-olds struggle and the birds succeed seems not to be given? The authors should also comment on this difference.

We have extended the paragraph comparing child and crow performance (line 551-560).

The authors have presented many different potential explanations for the findings in the various conditions. However, given the emphasis of self-control in the introduction, could the authors make a concluding statement in the discussion on what was learned with regard to the self-control abilities of the studied groups, if anything at all can be said with certainty?

References to self-control have been added in the conclusion.

It would be nice if the authors could make a bit clearer in the discussion that xyz was their main focus and abc is what they found. At the moment, the discussion is rather driven by what has been found, e.g., in lines 574 and following the results from the tool selection quality allocation condition are presented as the most prominent finding and for the first time the ability to process information is discussed, which doesn’t seem to be strong focus of the paper at all.

We have clarified the focus of the paper at the start of the discussion (line 524-527).

Minor issues:

- Line 50: comma before „and the quality” 

A comma has been added before “and the quality” (line 52).

- Line 51: I suggest to start a new paragraph for the sentence beginning with “One aspect that underlies”; that way, the first paragraph describes decision-making, whereas the second one focuses on self-control/delay of gratification, and will help the reader follow the structure and arguments of the paper.

We now start a new paragraph with “One aspect that underlies…” (line 54)

- Line 58: another good review article for the development of inhibitory control in children, which the authors could refer to, is: Garon, N., Bryson, S. E., & Smith, I. M. (2008). Executive Function in Preschoolers: A Review Using an Integrative Framework. Psychological Bulletin, 134(1), 31–60.

This reference has been added.

- Lines 64-67: The word “therefore” seems to be misplaced. “Therefore” seems to imply that the argument that decision-making involving tool use may require increased levels of complexity results from what has been described in the previous sentence; however, this is not the case. Rather, there seem to be two separate issues involved in decision-making involving tool use: 1) tool use requires delay of gratification and 2) tool use requires an increased level of relational complexity, as well as work-effort. Relatedly, the next sentence starting with “For example” doesn’t seem to be an example of what has been explained in the previous sentence. How is the need to pay attention to tool functionality linked to an increased relational complexity? I understand that paying attention to the tool per se (compared to only paying attention to the reward) increases relational complexity, but this seems to be regardless of whether or not attention is paid to tool functionality. Neither does the given example fit to the statement that tool use may require additional work-effort. I suggest that the authors rephrase this or make it clearer in how far their example represents either of the two issues mentioned in the previous sentence (increased relational complexity or additional work-effort).

This section has been rephrased (line 66 onwards).

- Lines 68/69: “has been found in various non-human species, including primates and birds” – the nomenclature is not quite correct here – the authors use the label “species” and when they continue with their examples (birds, primates) the phrasing seems to suggest that primates and birds are labels for species. Please try to be as scientifically correct as possible and try to rephrase.

This has been rephrased (line 71-73).

- Lines 77 and 79: “tool-making” – this is the first time the authors use the label “tool-making” – they started the paragraph differentiating between tool-using and non-tool-using animals and then there is the word “tool-making”. It doesn’t become clear whether these labels are used interchangeably (they shouldn’t). Is there a reason for the switch in the labels? If so, this has to be explained, as otherwise this leaves the reader potentially confused. If there is no reason, I suggest using just one label throughout the manuscript (presumable “(non-) tool-using” as the paper focuses on tool use, not on tool manufacture) in order to not introduce any ambiguity or confusion.

The paragraph was restructured to define the terms tool-using and tool-making before referring to them (line 81-86).

- Line 84: please insert “;” after “i.e.”

This has been changed.

- Line 110: comma after “humans”?

- Line 144: comma after “branches”?

We did not use Oxford commas throughout the manuscript and have thus not added these commas.

- Line 122: “require” instead of “requires”?

- This has been changed.

- Line 170: remove comma in “box, was”

- This has been changed.

- Line 170: comma after “i.e.”

- This has been changed.

- Line 214: “Supplemental Video” – please specify which video (2 in this case)

This has been added.

- Lines 351/352: please provide references for the R packages 

- References have been added.

- Line 367: “data were” instead of “data was”

- This has been changed.

- Line 372: “in S1 Table” instead of “i S1 Table”

- This has been changed.

- Lines 377/378 and 425-427: replace “X” with “χ”

- This has been changed.

- Line 378 and 426-427: italicize “p”-values

- Line 378: insert space between “<” and “0.001”

- Lines 378/379 and 426-427: remove the first zero in “0.001” – as the value is bound between 0 and 1, there does not need to be a 0 before the decimal point 

- Table 1: remove zeros before the decimal points

These changes to the p-values have been made throughout the results section.

- Table 1: It is not explained what “S1&2” stands for

- This has been added.

- Lines 387-392: “For the motivation condition, the correct choice was the immediately available most preferred reward over the functional tool, with the same most preferred reward inside the apparatus. For the quality allocation condition, the correct choice was the immediately available most preferred reward over the tool when the least preferred reward was inside the apparatus, or the tool over the least preferred reward, when the most preferred reward was inside the apparatus.” This was already explained in the methods section and it’s unclear why this is repeated here. Delete.

- This section has been removed.

- Table 2: 

o please remove the zero before the decimal point all in p-values. 

o Insert a space between “<” and the respective p-values

o The authors write “<.0001” several times. The convention is to use .001 (3 decimal places and the authors should consider changing this.

o In the column “tool functionality”, “P” should be “p”

These changes have been made.

o Why are some p-values italicized? This is not explained in the figure caption and needs to be added. 

We have now italicized all p-values throughout the manuscript. 

- Line 451: “-“ after “3” and “4” 

- This has been changed.

- Line 457: “-“ after “3”

- This has been changed.

- Lines 463-464: This heading is too vague. Isn’t this section more about a comparison? The performance of the crows and the humans was already presented in the previous sections, so labelling this section “performance in NC crows, …children, and adult humans” is potentially confusing. I suggest rephrasing the heading so that it is more specific and describes exactly what the following section is about 

- This heading has been changed to “Comparison with previous studies in Goffin’s cockatoos and orangutans”

- Lines 456-466: “we illustrate the performance…with that” – this seems grammatically incorrect. Better: we compare performance…with that?

- This has been changed.

- Line 497: remove comma after “humans”

- This has been changed.

- Line 519: “succeed” instead of “success”

- This has been changed.

- Line 526: “species” instead of “species’”

- This has been changed.

- Line 539: comma after “mentality”

- This has been added

- Line 584: “three” instead of “3”

- This has been changed.

- Line 684: the “S” in “Psychological science” should be capital letter

- Supplementary Material S3 and S4 Table: In the caption of both tables, replace “models” with “model”

- This has been changed.

---

## [Decision Letter · Decision Letter 1]

12 Feb 2020

Decision-making flexibility in New Caledonian crows, young children and adult humans in a multi-dimensional tool-use task

PONE-D-19-18492R1

Dear Dr. Frohnwieser,

We are pleased to inform you that your manuscript has been judged scientifically suitable for publication and will be formally accepted for publication once it complies with all outstanding technical requirements.

As you will see there are some minor comments from one of the reviewers regarding some typos etc. I think your manuscript will benefit if you attend to these comments.

With kind regards,

Juliane Kaminski

Academic Editor

PLOS ONE

Additional Editor Comments (optional):

Reviewers' comments:

Reviewer's Responses to Questions

**Comments to the Author**

1. If the authors have adequately addressed your comments raised in a previous round of review and you feel that this manuscript is now acceptable for publication, you may indicate that here to bypass the “Comments to the Author” section, enter your conflict of interest statement in the “Confidential to Editor” section, and submit your "Accept" recommendation.

Reviewer #1: All comments have been addressed

Reviewer #2: All comments have been addressed

2. Is the manuscript technically sound, and do the data support the conclusions?

Reviewer #1: Yes

Reviewer #2: Yes

3. Has the statistical analysis been performed appropriately and rigorously? 

Reviewer #1: Yes

Reviewer #2: Yes

4. Have the authors made all data underlying the findings in their manuscript fully available?

Reviewer #1: Yes

Reviewer #2: Yes

5. Is the manuscript presented in an intelligible fashion and written in standard English?

Reviewer #1: Yes

Reviewer #2: Yes

6. Review Comments to the Author

Reviewer #1: The authors have addressed all my previous comments. I recommend publication pending some minor adjustments (see below):

Line 222: "The ‘stick-apparatus’ for the humans was like [50]": consider rephrasing this sentence.

Lines 417 and 423: “we included the random effect of subject 1+Trial+Condition|ID”: I would suggest something along the following lines: “we included the random effect of subject ID and the random slopes of trial and condition within subject ID”

Lines 422 and 428: “random effect” (not “random factor”)

Line 502: why is a z statistic reported here (it does not match the z-value in Table S4). Should probably be the chi sqared value.

Table S4-1 is insufficiently described and labelled. I did not understand which comparisons were made in the different rows.

Table S3-1: layout could be improved.

Table S3 and S4: empty tables are still included

Reviewer #2: The authors address all comments in a satisfying manner and I recommend this manuscript for publication. I have a few remaining minor issues (line numbers relate to the track-changed manuscript):

Introduction, line 135: “decisions in across five conditions” – delete either “in” or “across”

Introduction, line 146: “each require” – not entirely sure about this, but the word “each” seems to imply singular, therefore “require” should be “requires”?

Introduction, lines 149-152: “delayed gratification and tool functionality understanding appear to develop in children at different ages [45, 46], but have not previously been tested simultaneously. Thus, we expected children’s ability to solve these tasks to increase with age.” The second sentence (stating that an increase in task performance is expected with age) does not necessarily follow from the first (that delayed gratification and tool functionality understanding start developing at different ages). In order for this conclusion to make sense, the authors would need to add that while delayed gratification and tool functionality understanding emerge at different ages, there is an increase in these skills over childhood.

Methods, line 219: “two different “stick apparatus” – “apparatus” needs to be changed to plural, “apparatuses”

Methods, line 279 and 286: the authors use “step 1” in the former and “Step 1” in the latter sentence – needs to be consistent

Methods, line 394: “received the following order of” – maybe “following” not needed as the order is given in the same sentence?

Analysis, lines 423 and 428: The random effect for child is once labelled “ID” and once “individual” – should be consistent

Analysis, line 452: “for multiple comparison” – “comparison” needs to be changed to plural “comparisons”

7. PLOS authors have the option to publish the peer review history of their article (what does this mean?). If published, this will include your full peer review and any attached files.

Reviewer #1: No

Reviewer #2: No

---

## [Editor Report · Acceptance letter]

21 Feb 2020

PONE-D-19-18492R1 

Decision-making flexibility in New Caledonian crows, young children and adult humans in a multi-dimensional tool-use task 

Dear Dr. Frohnwieser:

I am pleased to inform you that your manuscript has been deemed suitable for publication in PLOS ONE. Congratulations! Your manuscript is now with our production department. 

With kind regards,

on behalf of

Dr. Juliane Kaminski 

Academic Editor

PLOS ONE